# Sphingosine-1-phosphate suppresses GLUT activity through PP2A and counteracts hyperglycemia in diabetic red blood cells

Nadine Thomas [1,7], Nathalie H. Schröder [1,7], Melissa K. Nowak [1], Philipp Wollnitzke[1], Shahrooz Ghaderi[1], Karin von Wnuck Lipinski[1], Annalena Wille [1], Jennifer Deister-Jonas [1], Jens Vogt[1], Markus H. Gräler [2,3], Lisa Dannenberg[4], Tobias Buschmann [1], Philipp Westhoff [5], Amin Polzin[4], Malte Kelm[4], Petra Keul [1], Sarah Weske[1] & Bodo Levkau [1,6] ✉

Red blood cells (RBC) are the major carriers of sphingosine-1-phosphate (S1P) in blood. Here we show that variations in RBC S1P content achieved by altering S1P synthesis and transport by genetic and pharmacological means regulate glucose uptake and metabolic flux. This is due to S1P-mediated activation of the catalytic protein phosphatase 2 (PP2A) subunit leading to reduction of cell-surface glucose transporters (GLUTs). The mechanism dynamically responds to metabolic cues from the environment by increasing S1P synthesis, enhancing PP2A activity, reducing GLUT phosphorylation and localization, and diminishing glucose uptake in RBC from diabetic mice and humans. Functionally, it protects RBC against lipid peroxidation in hyperglycemia and diabetes by activating the pentose phosphate pathway. Proof of concept is provided by the resistance of mice lacking the S1P exporter MFSD2B to diabetes-induced HbA1c elevation and thiobarbituric acid reactive substances (TBARS) generation in diabetic RBC. This mechanism responds to pharmacological S1P analogues such as fingolimod and may be functional in other insulin-independent tissues making it a promising therapeutic target.

Red blood cells (RBC) import glucose through the superfamily of 14 related GLUT sugar transporters that are characterized by moderate binding affinity and rapid transport rate along a concentration gradient[1,2]. This occurs in an insulin-independent, satiable and passive way[3]. Human RBC express the highest levels of the facilitative glucose transporter 1 (GLUT1) of all cell types, whereas murine RBC undergo a postnatal switch from GLUT1 to GLUT4[4]. Since mature RBCs have neither cell nucleus nor mitochondria[5], they depend on glucose as the sole energy source. In addition to ATP production through direct glycolysis, the pentose phosphate pathway (PPP) shunts glucose in a parallel metabolic pathway to yield NADH as powerful reducing agent utilized to maintain redox balance and prevent aberrant oxidation[6]. Accordingly, defects in glucose transport activity are associated with impaired RBC morphology, reduced deformability and shortened lifespan[7,8].

Cell membrane resident GLUT1 builds noncovalent homodimers and tetramers[9] and forms clusters as active working units[10]. The majority of GLUT clusters are regulated by lipid rafts and restricted in

[1]Institute of Molecular Medicine III, Heinrich Heine University, Düsseldorf, Germany. [2]Department of Anesthesiology and Intensive Care Medicine, Center for Sepsis Control and Care, Jena University Hospital, Jena, Germany. [3]Center for Molecular Biomedicine, Jena University Hospital, Jena, Germany. [4]Division of Cardiology, Pulmonology, and Vascular Medicine, University Hospital Düsseldorf, Düsseldorf, Germany. [5]Institute of Plant Biochemistry, Cluster of Excellence on Plant Sciences (CEPLAS), Heinrich Heine University, Düsseldorf, Germany. [6]CARID, Cardiovascular Research Institute Düsseldorf, Medical Faculty and University Hospital, Düsseldorf, Germany. [7]These authors contributed equally: Nadine Thomas, Nathalie H. Schröder. ✉e-mail: Bodo.Levkau@med.uni-duesseldorf.de

size by the cytoskeleton and through glycosylation[10]. Lipid rafts or detergent-resistant membranes are membrane domains containing high levels of cholesterol, sphingolipids, and specific proteins crucially involved in cell signaling and protein assembly[11]. The spatial recruitment and clustering of proteins and lipids in lipid rafts regulates numerous signaling and protein transfer processes as shown for insulin receptors, integrins, and T cell antigen receptors[12,13]. Both GLUT1 and GLUT4 have been found in detergent-resistant membranes by biochemical means[14,15], and spatial association of GLUT1 with lipid rafts has been confirmed by super-resolution imaging[10]. However, the majority of these studies have been conducted in transformed cell lines, where GLUT1 has been shown to translocate from intracellular storage pools to the cell surface upon metabolic stress[16,17], or where randomly distributed GLUT transporters were recruited to lipid rafts in response to glucose deprivation[16,17]. In contrast, only little information is available on the physiological, insulin-independent regulation of GLUT1 and GLUT4 activity and trafficking in RBC.

Lipid rafts are highly dynamic in their physicochemical and biological properties suggesting that their composition may affect GLUT function. In fact, the phospholipid composition of the membrane has been shown to have dramatic effects on GLUT4 function as measured by the $k_{cat}$ of glucose transport: liposomes containing anionic phospholipids such as phosphatidic acid, phosphatidylserine and phosphatidylinositol stabilized and activated GLUT4, whereas liposomes containing conical lipids such as phosphatidylethanolamine and diacylglycerol enhanced transporter activity but without stabilizing protein structure[18]. Thus modulation of GLUT activity by alterations in plasma membrane phospholipid content that take place under physiological and pathophysiological settings may be a still underexplored metabolic regulatory mechanism. In this respect, sphingosine-1-phosphate (S1P) production was shown to increase in human RBC at high altitude (>5000 m) and in murine RBC under chronic hypoxia, respectively[19]. In consequence, S1P stimulated the anchoring of deoxygenated hemoglobin to RBC membranes thereby releasing membrane-bound glycolytic enzymes. The subsequent increase in glycolysis resulted in higher 2,3-bisphosphoglycerate (2,3-BPG) levels thereby promoting $O_2$ release in response to chronic hypoxia[19]. Importantly, these effects were mediated by intracellular and/or membrane-associated S1P and not through S1P receptor signaling.

In our study, we hypothesized that alterations of intracellular S1P concentration in RBC affects glucose import through GLUTs. We demonstrate that RBC S1P content regulates GLUT1 activity in human RBC and GLUT4 activity in murine RBC, respectively, and that modulating S1P synthesis by sphingosine kinase 1 (Sphk1) and S1P export through the Major Facilitator Superfamily Domain Containing 2B (Mfsd2b), respectively, dynamically regulates glucose uptake and glycolytic flux. We identify the molecular mechanism by which intracellular S1P dynamically regulates GLUT activity and provide evidence that RBC employ it as safeguard against pathophysiological metabolic conditions such as hyperglycemia in human and murine diabetes mellitus.

## Results

### Alterations of RBC S1P control glucose uptake and glycolysis

To investigate whether there was a causal relationship between intracellular S1P levels and RBC glucose uptake, we either pharmacologically raised or lowered intracellular S1P, respectively, and measured the corresponding glucose uptake rate. Firstly, we used D-erythro-sphingosine loading as an established method to increase intracellular S1P in RBC because of their high sphingosine kinase activity and lack of S1P degrading enzymes. Secondly, we employed serum albumin as an extracellular S1P acceptor to subsequently unload them from intracellular S1P[20]. Incubation with sphingosine resulted in a 15-fold increase in intracellular S1P concentration (~2 pmol/Mio RBC) (Fig. 1A) due to the highly efficient conversion of

sphingosine to S1P (Supplementary Fig. 1A). Interestingly, this treatment was accompanied by a 34% reduction in glucose uptake rate (Fig. 1C). Conversely, incubation of S1P-loaded RBC with albumin reduced intracellular S1P concentration by 40% due to efficient S1P efflux (Fig. 1B). This was accompanied by restoration of RBC glucose uptake rate back to normal (Fig. 1C). No sphingosine efflux was observed (Supplementary Fig. 1B). To ensure that the effect was, indeed, due specifically to S1P, we extracted S1P from RBC using the S1P-neutralizing antibody Sphingomab as previously described[21]. Sphingomab but not an isotype-matched control IgGt efficiently extracted 0.8 pmol S1P per $10^6$ sphingosine-loaded RBC (Fig. 1D) and increased glucose uptake similar to albumin (Fig. 1E). Using the only antibody available worldwide that reliably detects an extracellular domain of GLUT4[22], we have observed that sphingosine led to a clear reduction of GLUT4 on the RBC surface (Fig. 1F).

To explore the physiological consequences of the reduction on glucose uptake, we performed metabolic flux experiments using 1,2,3-$^{13}C_3$-D-glucose. The triple $^{13}C$ label on position 1, 2 and 3 of the glucose isotope allows to distinguish the lactate production from direct ($^{13}C_3$ lactate) glycolysis from that following loss of the first carbon atom $^{13}C_1$ in the oxidative part of the pentose phosphate pathway (PPP) resulting in $^{13}C_2$ lactate generation. A reliable readout for the predominance of one over the other is provided by the $^{13}C_3$ lactate/$^{13}C_2$ lactate ratio. Interestingly, we observed a 50% increase in the $^{13}C_3$ lactate/$^{13}C_2$ lactate ratio (Fig. 1H) suggesting that increased glycolysis may compensate for less glucose entry. The preferential flux to direct glycolysis instead of PPP was convincingly confirmed by measurements of the direct PPP metabolite phosphoribosyl pyrophosphate (PRPP) as the relative exchange of $^{13}C_2$ PRPP was reduced by 400% (Fig. 1H).

We also attempted S1P loading by direct supplementation of S1P instead of the detour through sphingosine. However, this was 5-fold less efficient (Supplementary Fig. 2A), did not alter glucose uptake (Supplementary Fig. 2B) and led to hemolysis from >4 μM on (Supplementary Fig. 2C). As sphingosine was much more efficient and did not lead to hemolysis (Supplementary Fig. 2C) we continued to use it as a tool to efficiently raise intracellular S1P.

In a next step, we investigated whether RBC harboring in vivo endogenously high S1P levels also exhibited an altered glucose uptake. RBC from mice treated with the S1P lyase (Sgpl1) inhibitor 4-deoxypyridoxine (DOP)[23] had 2-fold higher S1P concentrations than controls (Fig. 2A) and featured a 20% reduction in glucose uptake rate (Fig. 2C). Unloading them of S1P with albumin reduced intracellular S1P concentration by ~70% due to S1P efflux (Fig. 2B) and this, again, led to an increase in glucose uptake rate (Fig. 2C). We confirmed this in Sgpl$^{vav-cre+}$ mice, where we deleted the S1P lyase in hematopoietic cells. There, RBC S1P levels were 1.8-fold higher compared to Sgpl$^{vav-Cre-}$ (Fig. 2D) and could be reduced by S1P unloading with albumin (Fig. 2E). Accordingly, Sgpl$^{vav+Cre+}$ RBC had a 26% lower glucose uptake rate compared to Sgpl$^{vav Cre-}$ that was restored by S1P unloading (Fig. 2F). To test if the opposite – a genetically determined low S1P content in RBC – stimulated glucose uptake, we performed the same experiments in mice lacking sphingosine kinase 1 (Sphk1). Sphk1$^{-/-}$ RBC exhibited 13-fold lower S1P levels compared to control RBC and no efflux (Fig. 2G, H), which coincided with a 34% increase in glucose uptake rate (Fig. 2I). In agreement with Sphk1 being the main kinase responsible for sphingosine conversion to S1P in RBC, Sphk1$^{-/-}$ RBC could not be S1P-loaded by incubation with sphingosine but accumulated sphingosine instead (Fig. 2J, K). We could confirm these experiments using the Sphk1 inhibitor PF543. The absence of S1P-loading translated into the absence of any effects on glucose uptake (Fig. 2I). In contrast to D-erythro-sphingosine, neither of its stereoisomers D,L-threo-dihydrosphingosine, N, N-dimethylsphingosine and L-threo-sphingosine suppressed glucose uptake (Fig. 2L) in line with them not being metabolized by Sphks[24]. In summary, several ways to modulate

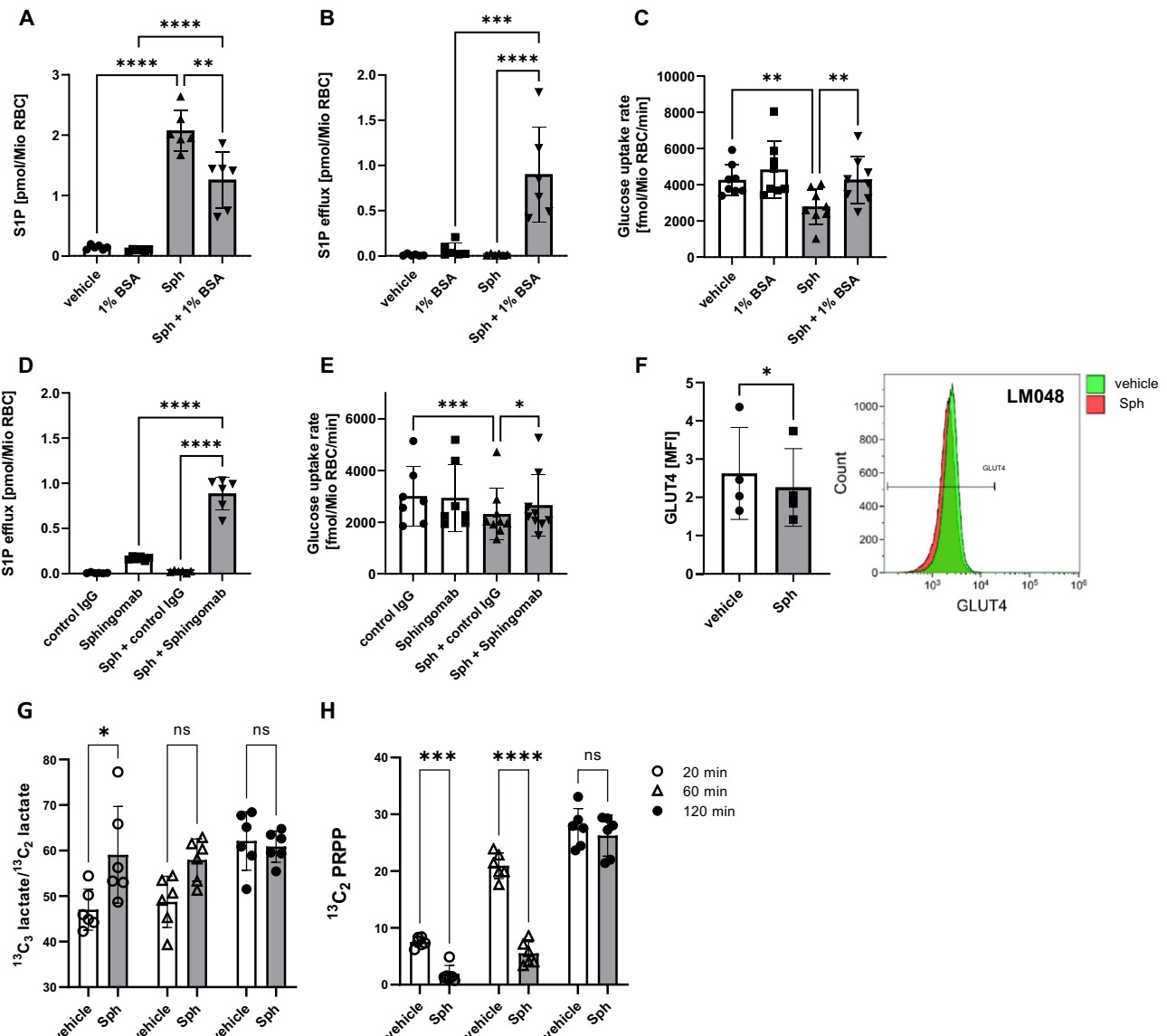

**Fig. 1 | S1P loading of murine RBC by sphingosine reduces glucose uptake, and S1P unloading by albumin or a S1P neutralizing antibody restores glucose uptake. A** C57Bl6 RBC S1P levels and (**B**) S1P efflux after incubation with 1 μM sphingosine (Sph) or vehicle (EtOH) for 30 min followed by incubation with or without 1% BSA for 20 min (n = 6 each). **C** Glucose uptake rate in RBC loaded or not with sphingosine and incubated subsequently with or without 1% BSA (n = 8). **D** S1P efflux from S1P or vehicle loaded RBC induced by incubation with 800 pmol/ml of the S1P neutralizing antibody Sphingomab or an isotype-matched control IgG1 (n = 6). **E** Glucose uptake rate in S1P-loaded RBC in the presence of Sphingomab or control IgG (n = 7/7/9/9). **F** GLUT4 cell surface localization without and with

sphingosine as determined by flow cytometry using the LM608 antibody that recognizes a strictly extracellular GLUT4 domain (n = 4). **G** $^{13}C_3$ lactate/$^{13}C_2$ lactate ratio, and (**H**) $^{13}C_2$ PRPP relative exchange rate in metabolic flux experiments using 1,2,3-$^{13}C_3$-D-glucose (n = 6 each). 1 pmol S1P/Mio RBC equals a concentration of 21 μMol/L S1P as calculated based on a mean MCV of 47.5 fl. The 'n' in the A-E refers to individual mice. Data are presented as mean ± sd and tested with paired one-way ANOVA followed by a Tukey's multiple comparison test (**A**–**E**), a paired two-tailed *t* test (**F**) or stack matched two-way ANOVA (G, H); ns= not significant; p** < 0.01; p*** < 0.001; p**** < 0.0001.

experimentally S1P concentration in RBC in vitro and in vivo dynamically affected glucose uptake rate.

## The S1P transporter Mfsd2b regulates glucose uptake in RBC

Mfsd2b belongs to the same Major Facilitator Superfamily as GLUTs 1–5 and has recently been identified as the main S1P exporter in RBC[20,25]. To test whether Mfsd2b was involved in the regulation of glucose uptake, we performed sphingosine loading and albumin unloading, respectively, with RBC from Mfsd2b$^{-/-}$ and Mfsd2b$^{+/+}$ mice. Even without loading, Mfsd2b$^{-/-}$ RBC had 40-fold higher S1P levels compared to Mfsd2b$^{+/+}$ RBC and exhibited a 20% decrease in glucose uptake rate (Fig. 3A, C). Mfsd2b$^{-/-}$ and Mfsd2b$^{+/+}$ could be further loaded with S1P, and this led to a further 30% decrease in glucose

uptake in Mfsd2b$^{-/-}$ RBC (Fig. 3C). Importantly, Mfsd2b$^{-/-}$ RBC could not be unloaded from S1P by albumin and, accordingly, exhibited no increase in glucose uptake rate (Fig. 3B, C). This mechanism was also present in nucleated cells. HEK293-Sphk1 cells exposed to sphingosine exhibited high intracellular S1P concentrations and reduced glucose uptake rate (Fig. 3D, F). As Mfsd2b was virtually absent there (Supplementary Fig. 3), albumin caused neither S1P efflux nor glucose uptake (Fig. 3E, F). However, Mfsd2b overexpression initiated S1P efflux and stimulated glucose uptake (Fig. 3F).

## FTY720-P but not FTY720 inhibits glucose uptake

FTY720 (fingolimod) is an approved prodrug for relapsing multiple sclerosis with its phosphorylated active form FTY720-P serving as a

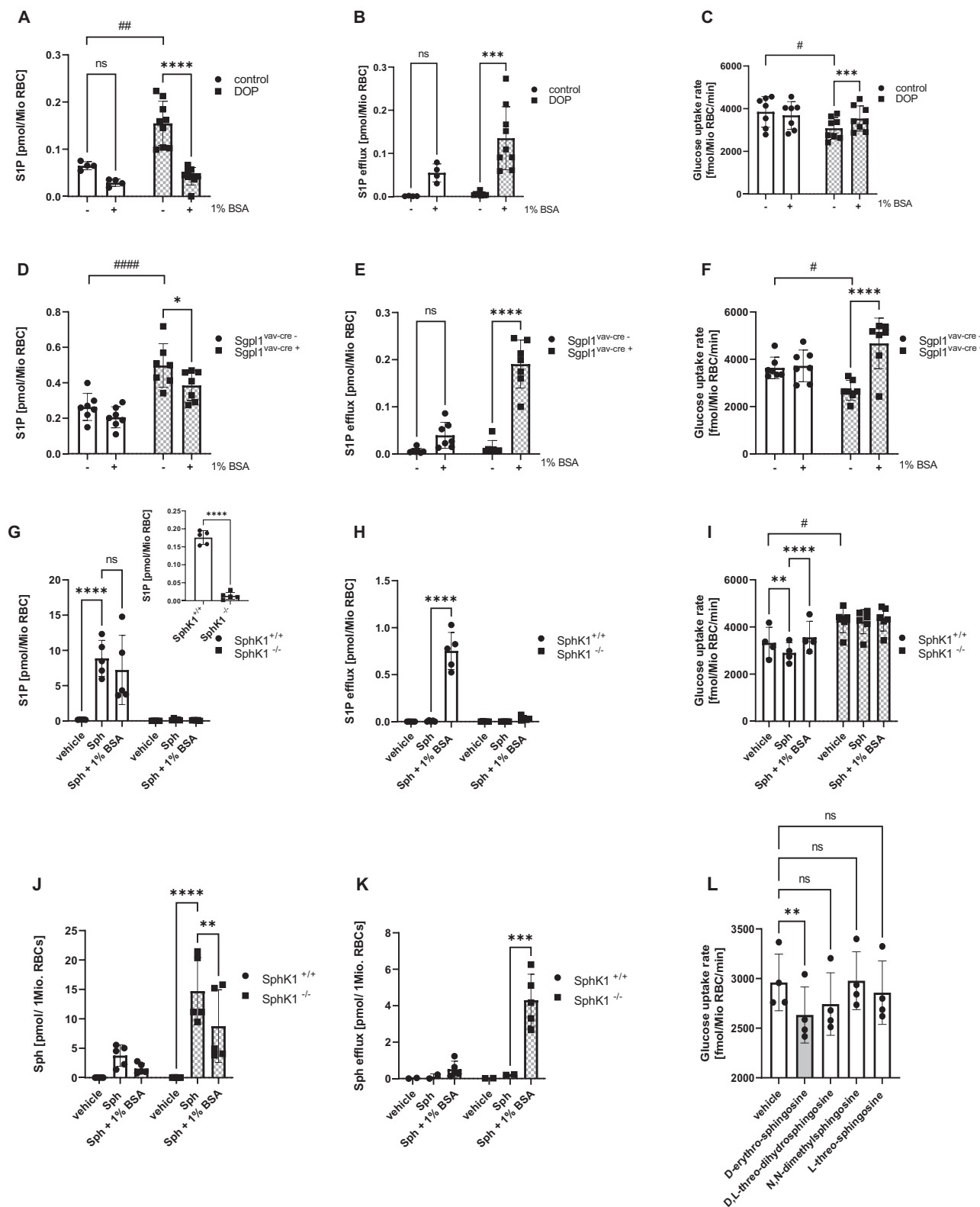

structural analog of S1P[26,27]. Thus we investigated whether FTY720 and FTY720-P have any influence on glucose uptake in RBC. Incubation of RBC with 1 μM FTY720 led to its accumulation but not to conversion to FTY720-P (Fig. 4A). This is in line with the lack of Sphk2 in RBC. Incubation with 1 μM FTY720-P also led to its accumulation in RBC (Fig. 4D) albeit at 30-fold lower levels than FTY720 in line with its polar nature. Only FTY720-P and not FTY720 suppressed glucose uptake (Fig. 4C, F). Despite the ability of albumin to unload RBC of FTY720

and FTY720-P (Fig. 4B, E) only FTY720-P but not FTY720 unloading restored glucose uptake (Fig. 4C, F). Interestingly, the efflux of both substances was not Mfsd2b dependent as it did not differ between Mfsd2b[+/+] and Mfsd2b[−/−] RBC (Fig. 4G, H).

## S1P regulates glucose uptake by modulating PP2A activity
Studies in tumor cell lines have shown that synthetic sphingolipids such as C2 ceramide block nutrient access by disrupting membrane

**Fig. 2 | Intracellular S1P concentration regulates RBC glucose uptake in vivo.**
**A** Intracellular S1P levels and (**B**) corresponding S1P efflux to BSA in RBC (isolated from mice treated or not with 4-deoxypyridoxine (DOP) for 3 weeks) (control n = 4; DOP n = 9). **C** Glucose uptake rate of same RBC before and after S1P unloading with BSA (n = 7). **D** Intracellular S1P levels and (**E**) corresponding S1P efflux in RBC from Sgpl1$^{vav-cre-}$ and Sgpl1$^{vav-cre+}$ mice (n = 7). **F** Glucose uptake of RBC from same mice as in (**E**) in the presence and absence of BSA for 30 min (n = 7 each). **G** intracellular S1P levels and (**H**) corresponding S1P efflux in RBC from SphK1$^{+/+}$ and SphK1$^{-/-}$ mice incubated with vehicle or 1 µM Sph followed by incubation without or with BSA; inset shows basal intracellular S1P level of SphK1$^{+/+}$ and SphK1$^{-/-}$ mice (n = 5 each). **I** Glucose uptake in RBC from SphK1$^{+/+}$ and SphK1$^{-/-}$ mice treated as in (**A**) (n = 4/6).

**J, K** sphingosine levels from (**G**) and (**H**), n = 5 each. **L** Glucose uptake rate in mouse RBC loaded with vehicle, 1 µM Sph, 1 µM D, L-threo-dihydrosphingosine, 1 µM N, N-dimethylsphingosine or 1 µM L-threo-sphingosine (n = 4). Data are presented as mean ± sd and tested with stack matched two-way ANOVA followed by a Tukey's multiple comparison test (**A**–**K**; # for comparing vehicle or genotype vs. treatment or genotype; * for comparing treatments) or paired one-way ANOVA in (**L**), and a mixed model due to randomly missing values in (**K**); in (**B**, **E**), there is also a significance in the controls with and w/o BSA if a paired *t* test was used. The use of the stack matched two-way ANOVA emphasizes that more S1P can be extracted from DOP-RBC. ns= not significant; p# < 0.05; p## < 0.01; p#### < 0.0001; p* < 0.05; p** < 0.01; p*** < 0.001; p**** < 0.0001.

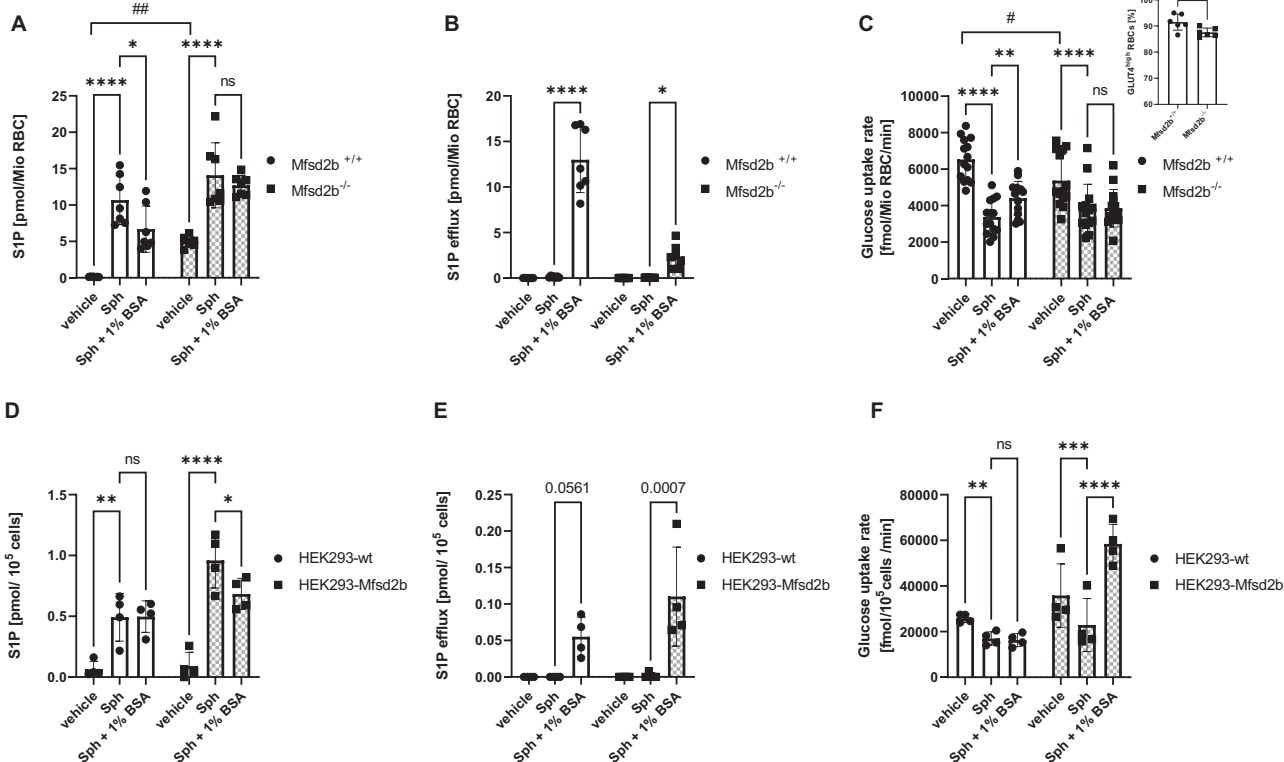

**Fig. 3 | Mfsd2b-mediated S1P efflux controls glucose uptake by modulating intracellular S1P. A** Intracellular S1P levels and (**B**) corresponding S1P efflux in RBC from Mfsd2b$^{+/+}$ and Mfsd2b$^{-/-}$ mice loaded or not with sphingosine and either S1P unloaded or not with BSA (n = 7 each). **C** Glucose uptake in RBC from Mfsd2b$^{+/+}$ and Mfsd2b$^{-/-}$ treated as in (**A**) (n = 14 each). Inset depicts GLUT4 cell surface localization. **D** Intracellular S1P levels and (**E**) corresponding S1P efflux in HEK293-SphK1-wt and HEK293-SphK1-Mfsd2b cells treated with vehicle or 1 µM Sph followed by

incubation without or with 1% BSA for 30 min (n = 4 each). **F** Glucose uptake rate in the same cells and treatments as in (**E**) (n = 4 each). Data are presented as mean ± sd and tested with stack matched two-way ANOVA (# for comparing Mfsd2b$^{+/+}$ vs. Mfsd2b$^{-/-}$; * for comparing treatments); in (**E**), there is a significance if a paired *t* test is used. ns= not significant; p# < 0.05; p## < 0.01; p* < 0.05; p** < 0.01; p*** < 0.001; p**** < 0.0001.

trafficking of nutrient transporters such as SLC2A1, SLC16A1, SLC16A3, SLC1A5 and SLC7A5 by activating the serine and threonine protein phosphatase 2A (PP2A)[28–30]. We thus tested whether S1P may regulate glucose uptake by affecting GLUT transporters through PP2A in RBC. Indeed, S1P loading of RBC with sphingosine clearly stimulated PP2A activity (Fig. 5A), and the concomitant reduction of glucose uptake and GLUT4 cell surface localization was prevented by the PP2A inhibitor ocadaic acid (Fig. 5B, C). Genetically low or high RBC S1P concentrations also coincided with low and high PP2A activity, respectively: SphK1$^{-/-}$ RBC featuring low S1P had low PP2A, whereas Mfsd2b$^{-/-}$ RBC featuring high S1P had high PP2A activity (Fig. 5C, D). To identify the underlying mechanism, we tested for direct effects of S1P on the native PP2A enzyme complex. For this, we immunoprecipitated PP2A from C57Bl6 RBC and added different sphingolipids and other compounds prior to measuring PP2A activity. Incubation with 1 µM S1P but not 1 µM sphingosine resulted in a 2-fold activation of PP2A (Fig. 5E). FTY720-P

(1 µM) but not FTY720 also activated PP2A (Fig. 5E), whereas C6 ceramide was effective at 15 µM as published[31] but not 1 µM (Fig. 5E). We then tested whether S1P affects not only the holoenzyme but also the activity of the PP2A catalytic subunit PPP2CA. Indeed, S1P and FTY720-P but not sphingosine, FTY720 or C6 ceramide activated human recombinant PPP2CA at 1 µM (Fig. 5F). These data identify S1P as a highly potent PP2A activator.

## S1P regulates GLUT1 function in human RBC
The same mechanism of S1P/PP2A-mediated regulation of glucose uptake was also present in human RBC: S1P-loading by sphingosine and unloading by albumin were similar to murine RBC (Fig. 6A, B) with the same consequences for glucose uptake (Fig. 6C) and PP2A activity (Fig. 6D). Using the receptor-binding domain of a recombinant envelope glycoprotein from the human T lymphotrophic virus fused to the enhanced green fluorescent protein coding sequence (H$_{RBD}$EGFP)

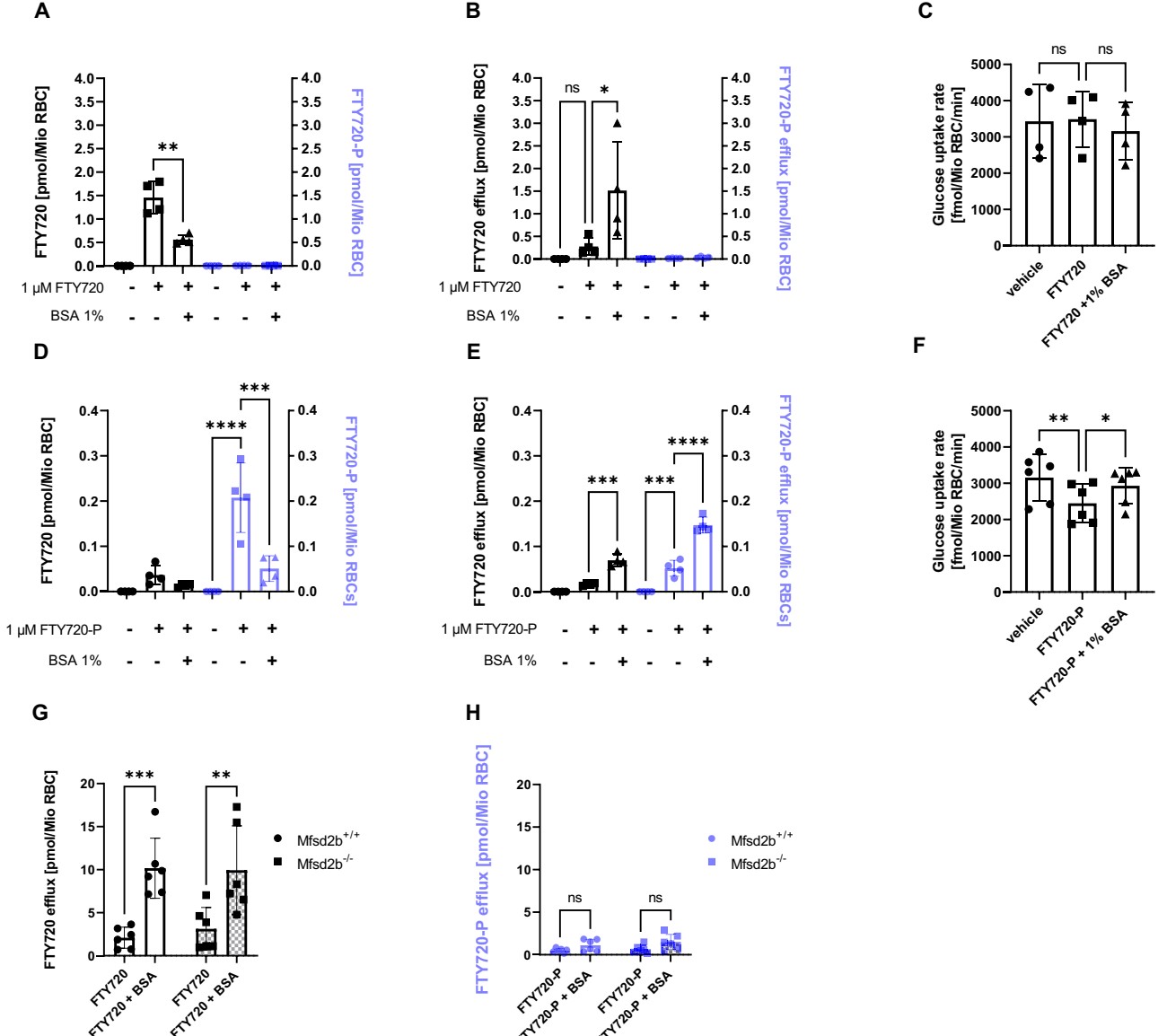

**Fig. 4 | FTY720 and FTY720-P transport into RBC occurs independently of Mfsd2b with FTY720-P but not FTY720 suppressing glucose uptake.**
**A** + **D** Intracellular FTY720 (black color) and FTY720-P (blue color) levels and corresponding FTY720 and FTY720-P efflux (**B** + **E**) after loading with 1 µM FTY720 (A/B) and 1 µM FTY720-P (D/E) followed by incubation without or with 1% BSA for 20 min (n = 4 each). **C** Glucose uptake of RBC treated with FTY720 and (**F**) FTY720-P, respectively, with and without addition of 1% BSA (n = 4 each).

Vehicle was EtOH for FTY720 and DMSO (0.1%) for FTY720-P. **G** Efflux of FTY720-loaded RBC (n = 6 each) and (**H**) FTY720-P-loaded RBC to 1% BSA in RBC from Mfsd2b[+/+] and Mfsd2b[−/−] mice (n = 6 each). The 'n' in the (**A**–**F**) refers to individual mice. Data are presented as mean ± sd and tested with stack matched one-way ANOVA followed by a Tukey's multiple comparison test; ns = not significant; p* < 0.05; P** < 0.01; P*** < 0.001; P**** < 0.0001.

as GLUT1 ligand[32], we observed that S1P-loading decreased GLUT1 localization on the RBC surface by ~50% (Fig. 6E, F). The rapid functional changes and GLUT1 localization kinetics may be related to rapid GLUT1 phosphorylation and dephosphorylation events known to affect GLUT1 cell surface translocation and glucose transport[33]. The most relevant phosphorylation site is Serine 226, phosphorylation of which is required for rapid increase in glucose uptake and enhancement of cell surface localization of GLUT1 after stimulation with the phorbol ester 12-O-tetradecanoyl-phorbol-13-acetate (TPA)[33]. We thus stimulated human RBC with TPA in the presence and absence of sphingosine and performed Western blotting with a phospho-Serine 226-specific GLUT1 antibody while simultaneously measuring glucose uptake. We observed that TPA lead to a clear increase in GLUT1 Serine 226 phosphorylation that was decreased by sphingosine (Fig. 6G). Sphingosine alone also lead to a decrease in GLUT1 phosphorylation

below baseline (Fig. 6G). Glucose uptake behaved similarly: it was clearly increased by TPA and suppressed by sphingosine (Fig. 6H).

## RBC S1P is higher in diabetic mice and patients
In RBC, the concentration of extracellular glucose determines the rate of glucose entry as their GLUTs transport glucose passively along concentration gradients[1,34]. In hyperglycemia associated in diabetes mellitus, hemoglobin is pathologically glycated as mirrored by increased HbA1c concentrations. We wondered whether RBC S1P concentrations are altered in diabetes, and what effect this may have on the parameters relevant to our study. For this, we made use of a murine model of hyperglycemia, HbA1C elevation and insulin-resistance (Fig. 7A, B) caused by a long-term (12 weeks) high caloric diet (diet-induced obesity; DIO). Interestingly, RBC from DIO mice exhibited 40% higher S1P concentrations, 53% increased PP2A activity

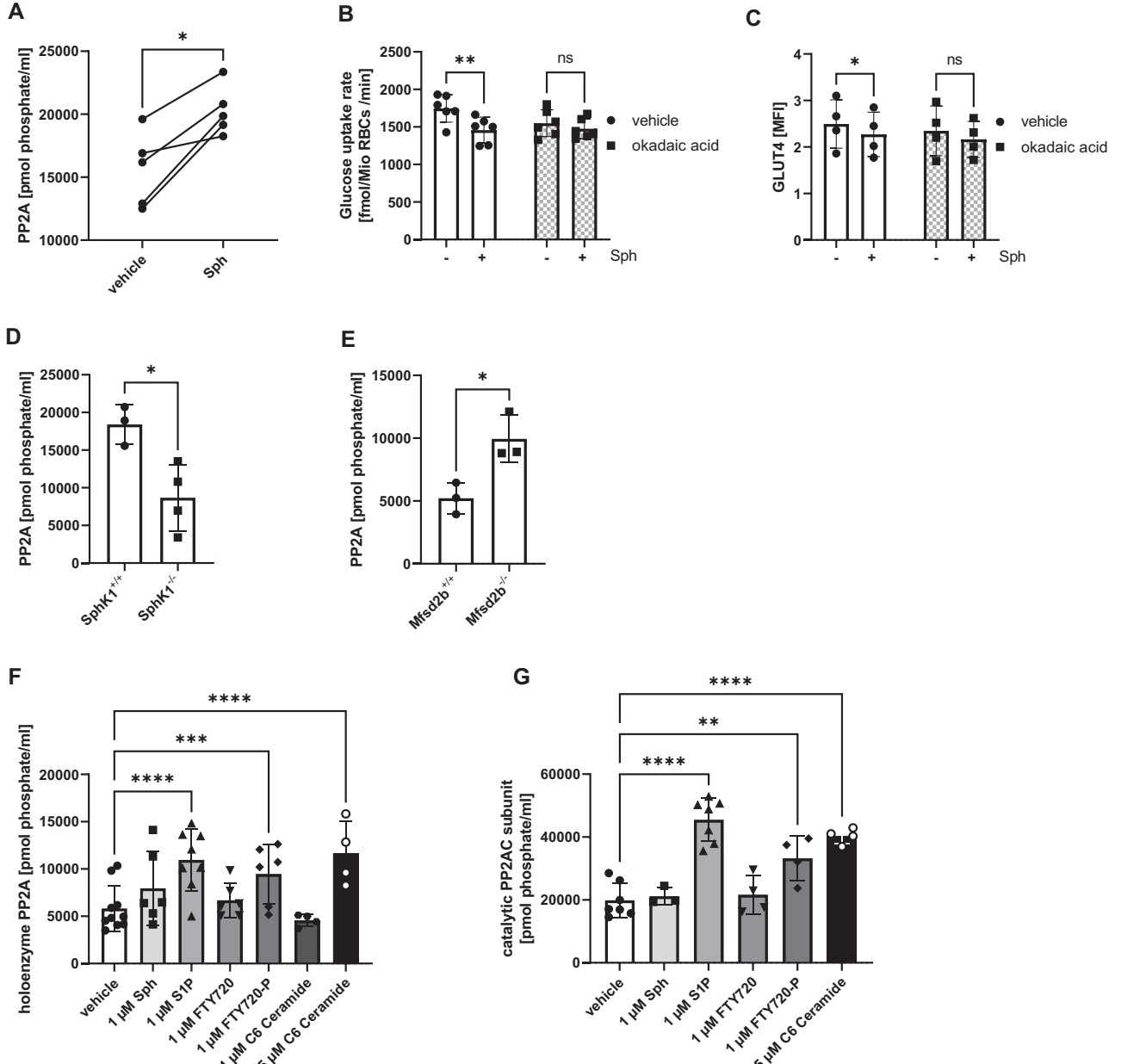

**Fig. 5 | S1P regulates glucose uptake by activating PP2A in vitro and in vivo.**
**A** PP2A activity measured after immunoprecipitation from RBC incubated with
vehicle and 1 μM Sph for 30 min (n = 5 each). **B** Glucose uptake and (**C**) GLUT4 cell
surface localization in sphingosine-loaded RBC in the absence or presence of 2 nM
of the PP2A inhibitor okadaic acid (n = 6 each). **D** PP2A activity measured in SphK1$^{+/+}$
and SphK1$^{-/-}$ RBC (n = 3/4), and (**E**) PP2A activity in Mfsd2b$^{+/+}$ and Mfsd2b$^{-/-}$ RBC
(n = 3 each). **F** PP2A activity of the holoenzyme immunoprecipitated from C57Bl6
RBC lysates and incubated consecutively with vehicle, sphingosine, S1P, FTY720,
FTY720-P and C6 ceramide in the indicated concentrations (n = 10/6/8/6/4/4).
**G** PP2A activity of human recombinant PP2AC catalytic subunit in the presence of
the same substances as in (**E**) (n = 7/3/7/4/4/4). The 'n' in the (**A**–**G**) refers to indi-
vidual mice. Data are presented as mean ± sd and tested with paired two-tailed *t* test
(**A**), two-way ANOVA (**B**, **C**), unpaired *t* test (**D**, **E**), and one-way ANOVA (**F**, **G**); ns=
not significant; p* < 0.05; P** < 0.01; P*** < 0.001; P**** < 0.0001.

and 29% decreased glucose uptake compared to lean controls
(Fig. 7C–E). They also exhibited an ~2-fold higher Sphk1 activity in
intact RBC as measured by C17-sphingosine conversion to C17-S1P
(Fig. 7F). This suggested that diabetes and/or hyperglycemia stimu-
lates S1P production and hence PP2A activity in RBC. These observa-
tions were also valid in a cohort of human patients with type 2 diabetes
mellitus (T2DM; patients' characteristics provided in Supplementary
table 1) with an average HbA1c of 8.9 mg/dl compared to 5.2 mg/dl in
case controls (Fig. 7G). T2DM RBC featured 34% higher S1P and 2-fold
higher PP2A activity (Fig. 7H, I), and these two parameters correlated

positively (Fig. 7K). This was accompanied by a 26% decrease in cell
surface GLUT1 (Fig. 7J).

**Mfsd2b deficiency averts diabetic RBC HbA1c elevation and lipid
peroxidation**
To monitor the effect of high RBC S1P on glucose uptake by diabetic
RBC over a long period, we fed Mfsd2b$^{+/+}$ and Mfsd2b$^{-/-}$ a DIO diet for 8
weeks. As result, both genotypes became obese and developed
a highly pathological but similar glucose intolerance as measured
by GTT (Fig. 8A). However, they dramatically differed in HbA1c

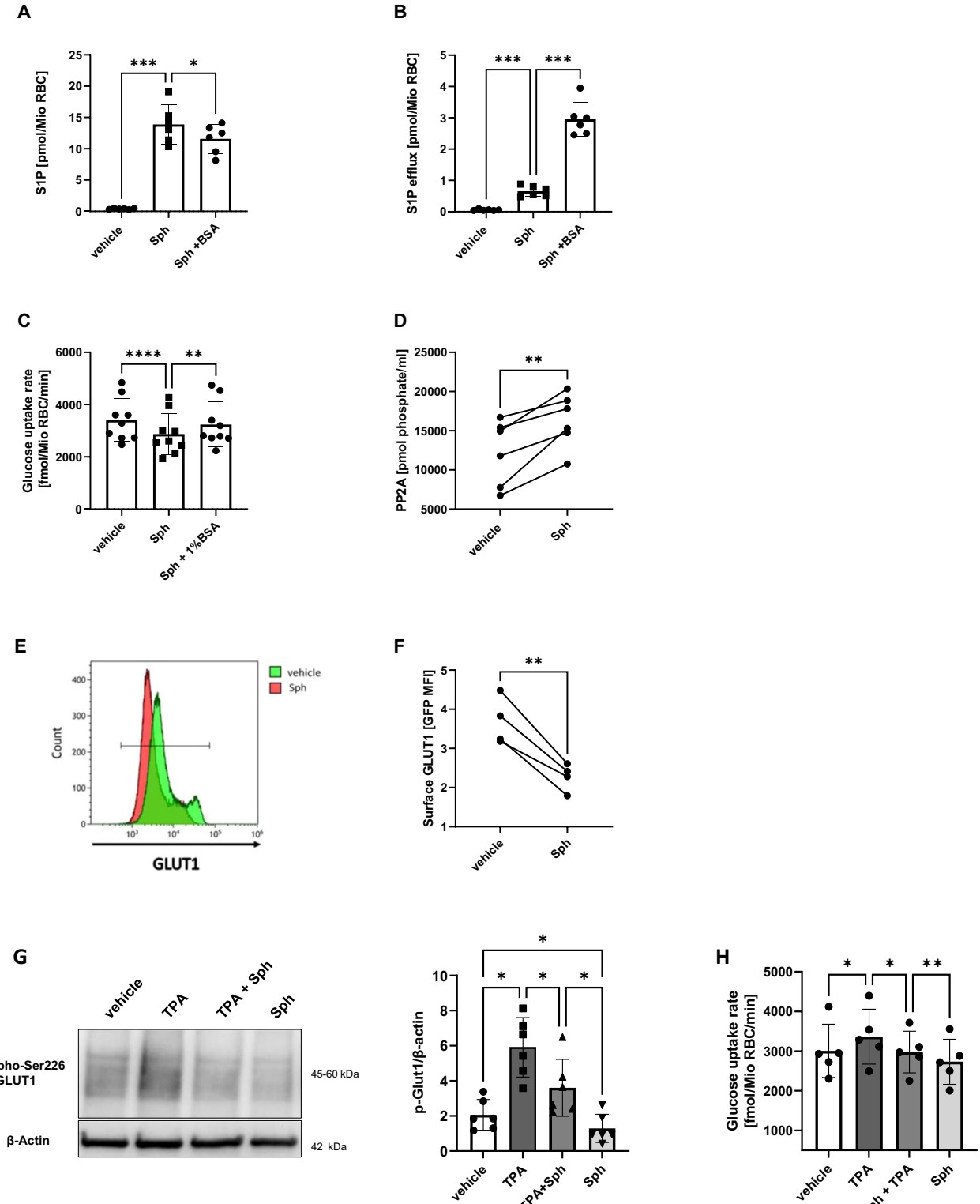

**Fig. 6 | S1P regulates human RBC glucose uptake, PP2A activity and GLUT1 phosphorylation and surface localization. A** Intracellular S1P levels and (**B**) corresponding S1P efflux in human RBC after incubation without or with 1 μM Sph for 30 min followed by exposure or not to 1% BSA for 30 min (n = 6 each). **C** Glucose uptake rate in human RBC treated as in (**A**) (n = 9). **D** PP2A activity of human RBC treated with 1 μM Sph or vehicle (n = 6 each). **E** Representative flow cytometry histograms of surface GLUT1 as measured by H$_{RBD}$EGFP binding after incubation with 1 μM Sph or vehicle for 30 min. **F** Quantification of GLUT1 fluorescence from human RBC treated with 1 μM Sph or vehicle (n = 4). MFI (mean fluorescence intensity). **G** Western blotting for GLUT1 Serine 226 phosphorylation (n = 6), and (**H**) glucose uptake rate in human RBC treated or not with 1 μM sphingosine for 20 min before a five-minute stimulation or not with 5 nM TPA at 37 °C (n = 5). The 'n' in the (**A**–**H**) refers to samples from individual patients. Data are presented as mean ± sd and tested with stack matched one-way ANOVA followed by Tukey's (**A**–**C**, **G**, **H**); paired two-tailed t test (**D**, **F**). p* < 0.05; p** < 0.01; p*** < 0.001; p**** < 0.0001.

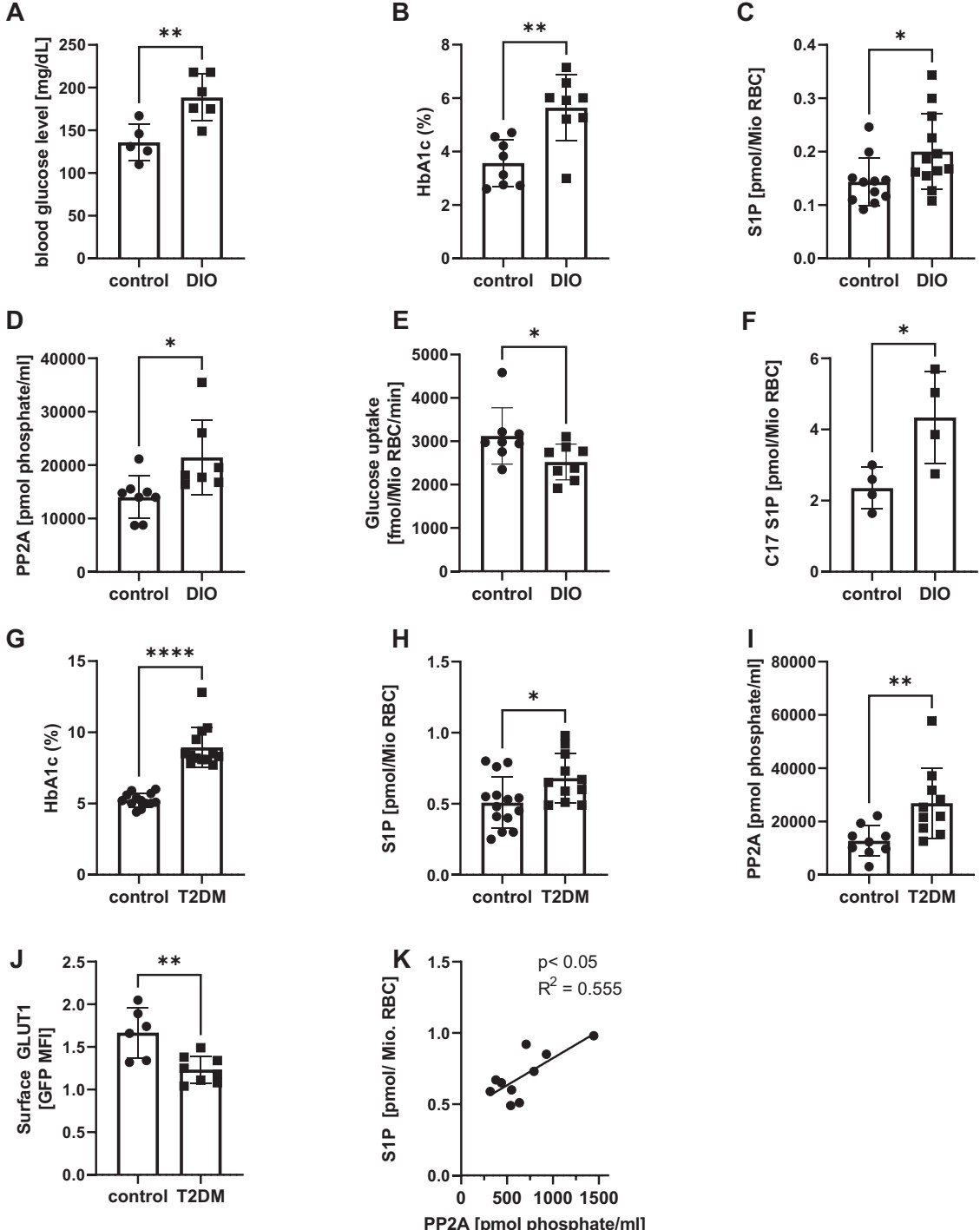

**Fig. 7 | RBC from chronically hyperglycemic mice and patients with diabetes mellitus type 2 have higher S1P and PP2A and lower GLUTs and glucose uptake.** C57Bl6 mice were fed either standard chow diet or high caloric diet (DIO diet) for 12 weeks and the following parameters were determined. **A** Fasting plasma blood glucose levels (n = 5/6), (**B**) HbA1c (n = 8/8), (**C**) S1P concentrations (n = 11/12), (**D**) PP2A activity (n = 8/7), (**E**) glucose uptake (n = 8 each), and (**F**) Sphk activity (n = 4 each) based on the conversion of C17-sphingosine to C17-S1P as measured by LC-MS/MS (n = 4). **G** HbA1c levels of human patients with and without T2DM (n = 13 case-controls, characteristics in Supplementary Table 1). **H** Intracellular S1P concentrations (n = 13 each), (**I**) PP2A activity (n = 10/9) and (**J**) cell surface GLUT1 (n = 8/6) on RBC from patients with and without T2DM (randomly selected from above groups). **K** Correlation between intracellular S1P and PP2A activity in RBC from T2DM patients (n = 10). Data are presented as mean ± sd and tested with two-tailed unpaid *t* test (**A**–**I**) and two tailed Pearson correlation coefficient (**J**); p* < 0.05; p**<0.01; p**** < 0.0001.

concentrations: while Mfsd2b$^{+/+}$ mice developed a 9-fold increase in HbA1c, Mfsd2b$^{-/-}$ mice were completely resistant to the HbA1c increase despite similar hyperglycemia (Fig. 8B). To understand whether the chronic reduction in intracellular glucose levels in Mfsd2b$^{-/-}$ DIO mice reflected by their normal HbA1c had any pathophysiological

consequences, we measured lipid peroxidation (TBARS) in RBC. The DIO diet lead to a clear increase in TBARS in Mfsd2b$^{+/+}$ RBC but to a strikingly lesser extent in Mfsd2b$^{-/-}$ RBC (Fig. 8C). This suggested that the high intracellular S1P in Mfsd2b$^{-/-}$ RBC had protected them against the lipid peroxidation caused by chronic hyperglycemia. Interestingly,

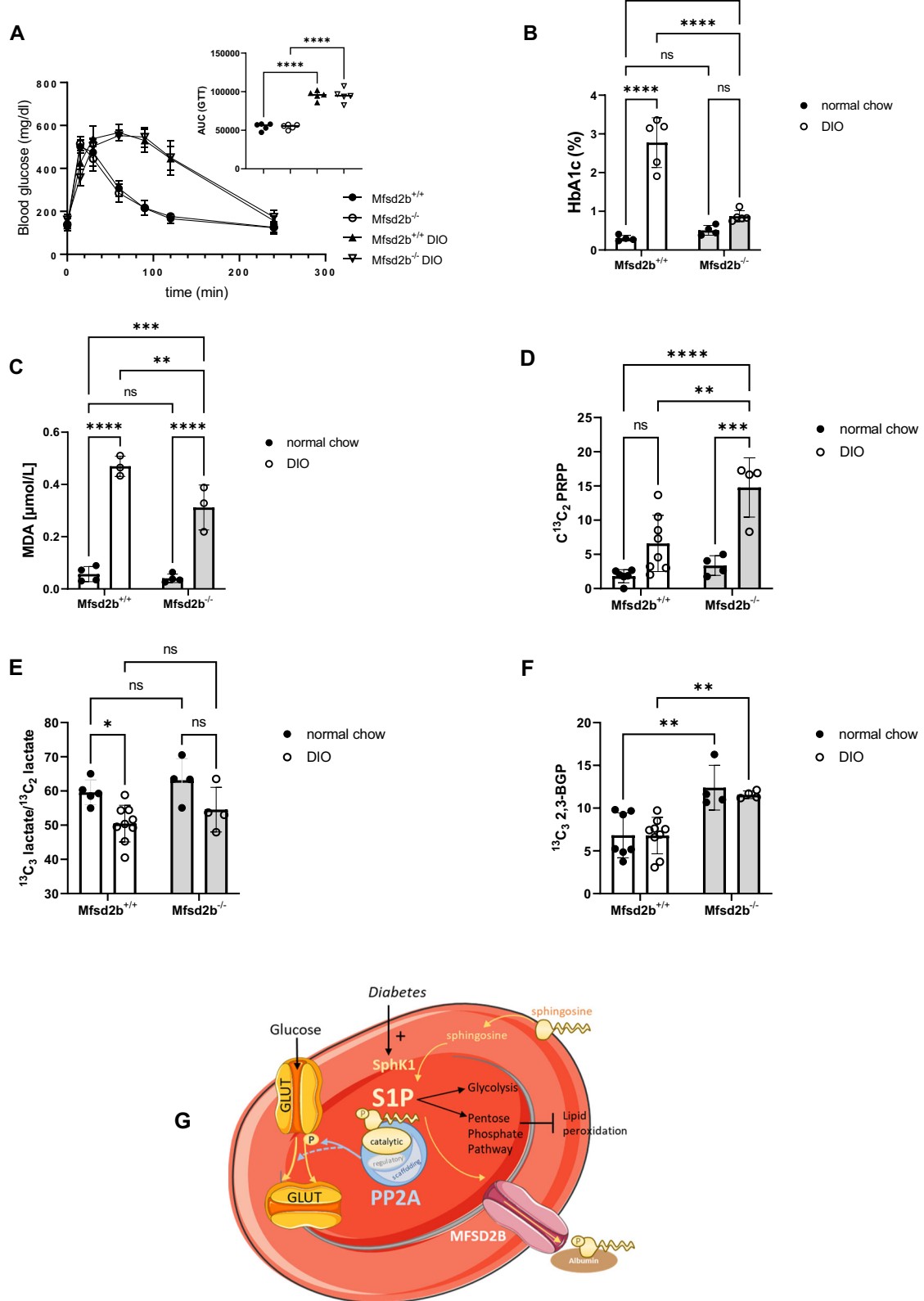

metabolic metabolic flux experiments using 1,2,3-$^{13}$C$_3$-D-glucose showed that the PPP metabolite $^{13}$C$_2$ PRPP was 2-3-fold higher in Mfsd2b$^{-/-}$ RBC compared to Mfsd2b$^{+/+}$ RBC suggesting a higher production rate of reducing equivalents (Fig. 8D) in line with lower lipid peroxidation. The $^{13}$C$_3$ lactate/$^{13}$C$_2$ lactate ratio was unchanged (Fig. 8E) indicating preserved glycolysis. We also measured 2,3-bisphosphoglycerate (2,3-BPG) and found it to be higher in Mfsd2b$^{-/-}$ RBC

than in Mfsd2b$^{+/+}$ RBC independently of diet (Fig. 8F) and also after sphingosine treatment (Supplementary Fig. 4).

## Discussion

This is, to our knowledge, the first report that RBC dynamically regulate intracellular S1P concentrations to modulate glucose uptake and metabolism in the context of diabetes. Previously, chronic hypoxia has been

**Fig. 8 | Chronically hyperglycemic Mfsd2b−/− mice are resistant to HbA1c elevation and lipid peroxidation and feature increased metabolic flux through PPP with unchanged glycolysis. A** Glucose tolerance test in Mfd2b+/+ and Mfsd2b−/− mice either on a normal chow diet or on a DIO diet for 8 weeks. Inset shows area under the curve (n = 5). **B** HbA1c (n = 5 each) and (**C**) TBARS levels (n = 4/3) in the same mice as determined by MDA measurements. **D** $^{13}C_2$ PRPP relative exchange rate in metabolic flux experiments using 1,2,3-$^{13}C_3$-D-glucose (20 min incubation); n = 6/8 Mfd2b+/+ and 4/4 Mfsd2b−/−. **E** $^{13}C_3$ lactate/$^{13}C_2$ lactate ratio in the same experiment (n = 5/9 Mfd2b+/+ and 4/4 Mfsd2b−/−). **F** $^{13}C_3$ 2,3-BGP relative exchange rate from the same experiment (n = 7/9 Mfd2b+/+ and 4/4 Mfsd2b−/−). Data are presented as mean ± sd and tested with one-way ANOVA (**A**) and ordinary two-way ANOVA followed by Tukey's (**B**–**F**); p* < 0.05; p** < 0.01; p**** < 0.0001. **G** Schematic of putative S1P functions in RBC: Blood glucose and sphingosine entry regulate Sphk1 activity and S1P production, whereas MFSD2B exports S1P, resulting in dynamic modulation of S1P levels in RBC. S1P associates with the catalytic PP2a subunit and activates the enzyme to reduce GLUT phosphorylation, cell surface localization and glucose uptake. The full arrow indicates dephosphorylation-dependent and the dotted arrow the dephosphorylation-independent PP2A-mediated GLUT internalization, respectively. Acute S1P elevation activates glycolysis to compensate for lower glucose entry at the expense of the pentose phosphate pathway (PPP), whereas chronically high S1P RBC levels as in Mfsd2b−/− RBC retain normal glycolysis but increase PPP metabolic fluxes to produce reducing equivalents for protection against hyperglycemia-induced lipid peroxidation.

shown to elevate RBC S1P for the purpose of stimulating the Rapoport-Luebering-Shunt to generate 2,3-BGP[19]. We, on the other hand, provide evidence that intercellular S1P governs the metabolically indispensable process of glucose influx by GLUTs in RBC under physiological and pathophysiological conditions. The mechanism we have identified is based on the ability of S1P to directly activate the catalytic subunit of PP2A. Synthetic sphingolipids have been shown to activate PP2A and kill glycolysis-dependent cancer cells in part by triggering the internalization of nutrient transporters and in part by mislocalizing PI(3,5)P2 generation by the lipid kinase PIKfyve which triggered cytosolic vacuolization and blocked lysosomal fusion[28]. Although both pathways depended on PP2A their nature had remained unknown perhaps due to the existence of different PP2A subsets[28] and existed side-by-side as the non-physiological short-chain C2 ceramide activated only the PP2A activity subset responsible for nutrient transporter loss and not for vacuolization[28,29]. Our data also suggest that S1P-activated PP2A may directly inhibit GLUT1 cell surface translocation by dephosphorylating Serine 226 in GLUT1, a post-translational modification known to inhibit GLUT1 cell surface translocation and glucose transport[33] (Fig. 8G). Although we have concentrated on RBC, our experiments with HEK293 cells overexpressing Sphk1 to raise intracellular S1P and Mfsd2b to get rid of it, respectively, with corresponding effects on glucose uptake suggest that this mechanism also functions in nucleated cells. This S1P metabolism-dependent regulation of glucose transport may thus be important in other tissues besides RBC with physiologically relevant insulin-independent GLUT activity such as the brain.

PP2A is a major tumor suppressor with decreased activity in several cancers that has been suggested as a promising target in cancer therapy[35]. This has led to the development of small molecule activators of PP2A (SMAPs) and heterocyclic activators of PP2A (iHAPs)[36]. While SMAPs and iHAPs stabilize the PP2A holoenzyme, natural sphingolipids such as C6 ceramide increase PP2A activity by binding to the catalytic subunit in stereospecific manner that requires the amide group, the primary hydroxyl group, and the secondary hydroxyl group of the sphingoid backbone[31]. We have identified S1P as the hereto most potent sphingolipid to directly activate PP2A (at least 10-fold more potent than C6 ceramide) and believe in a similar mode of action as ceramide[31] as it has the identical sphingoid backbone. Sphingoid bases and analoga such as sphingosine, FTY720 and DMS (but not FTY720-P or S1P) have been shown to activate PP2A by binding and displacing the PP2A inhibitors SET[37] and ANP32A[38] from the holoenzyme which relieves PP2A from inhibition. Here, we have identified another activation mechanism, where S1P and FTY720-P (but not sphingosine or FTY720) directly bind to and activate the catalytic subunit of PP2A in the absence of SET and ANP32A. In different cell types, these two mechanisms may be acting together or separately, and additional intermediate kinases as mediators may be involved. For example, PP2A has been shown to be activated in FTY720-treated endothelial cells[39], where either or both mechanisms may be acting – 'PP2A relief´ by FTY720 from SET and/or ANP32A resulting in indirect PP2A activation, and direct PP2A activation by FTY720-P after rapid phosphorylation of FTY720 by Sphk2. The only study that has described the opposite (S1P to inhibit PP2A)[40] ought to be interpreted with caution as PP2A should have but was not increased in erythrocyte-specific Sphk1 knockouts[40], and the 6 μM S1P used there in RBC induced hemolysis in our hands (Supplementary Fig. 2A).

In contrast to its downregulation in cancer, PP2A is hyperactivated in diabetes mellitus. In fact, the PP2A-dependent desensitization of the insulin/Akt signaling pathway has been proposed as pathophysiological principle in insulin resistance[41–43]. To our knowledge, there are no reports on the sphingolipid content or PP2A activity in diabetic RBC. Here, we have observed higher S1P levels and PP2A activity in RBC from human T2DM patients and mice with chronic hyperglycemia, respectively. Most importantly, we have demonstrated that high S1P (as in Mfsd2b deficient mice) completely prevented HbA1C elevation in hyperglycemia, clearly suggesting a protective role of S1P against RBC glucose overload. In this scenario, intracellular S1P increases in reaction to chronic hyperglycemia or hyperglycemia-associated phenomena and stimulates PP2A in a counter-regulatory manner to restrict glucose influx. This restriction would protect RBC against the many deleterious effects of intracellular hyperglycemia such as lipid peroxidation[44,45] as we have seen in our study. Interestingly, in contrast to the acute elevation of intracellular S1P after sphingosine exposure, metabolic flux through the PPP was clearly higher in DIO Mfsd2b−/− RBC compared to DIO Mfsd2b+/+ RBC, and it also increased more efficiently in Mfsd2b−/− than in Mfsd2b+/+ RBC under DIO suggesting a more effective production of reducing equivalents in line with better protection against lipid peroxidation (Fig. 8G). It is remarkable that the increase in PPP did not occur at the expense of glycolysis as the $^{13}C_3$ lactate/$^{13}C_2$ lactate ratio was unchanged. These data are directly relevant to a recent study on the effects of S1P on RBC storage quality, where S1P boosted glycolysis in stored human and murine RBCs at the expense of the PPP metabolic flux resulting in increased storage-induced oxidation[46]. This is in line with our data on the effects of acute S1P raising in RBC but curiously, not with the chronicall high S1P levels in Mfsd2b−/− resulting in increased PPP metabolic fluxes and anti-oxidative capacity but without hampering glycolysis; alternatively, a yet unknown metabolite is involved.

Diabetic RBC are the only tissue we have so far found this mechanism to have pathophysiological relevance for. In fact, there were no overall effects on hyperglycemia or glucose tolerance in Mfsd2b−/− mice despite the suppression of both Hb glycation and lipid peroxidation in RBC. Nevertheless, RBC protection against glycation and lipid peroxidation is generally assumed to be beneficial in diabetes. Further studies are needed to find out whether our findings may be relevant to chronic kidney disease[40] and sickle cell disease[47] as RBC pathologies associated with S1P perturbations. As to the nature of mechanism behind the increased RBC S1P content in diabetes, it may be due to the higher Sphk1 activity we have observed in diabetic RBC and perhaps to less active Mfsd2b restricting S1P efflux as its function is hampered by high intracellular pH[25], a characteristic feature of diabetic RBC[48].

In summary, we have identified a mechanism that dynamically regulates glucose uptake in RBC in response to metabolic cues from the

environment and does so by controlling S1P production and disposal (Fig. 8G). This mechanism may also be functional in other insulin-independent tissues and deserves future investigation. Finally, the idea of intracellular S1P fluctuations affecting PP2A activity may be relevant to a variety of human diseases beyond diabetes and could be exploited pharmacologically by already known drugs intervening in S1P homeostasis.

## Methods

### Human samples
Human blood was donated by healthy volunteers and control and diabetic patients after written informed consent without monetary compensation. The study was in accordance with the Declaration of Helsinki and was approved by the University of Düsseldorf Ethics Committee.

### Mouse models
All mouse experiments were approved by the LANUV (LANUV Recklinghausen, Germany) as stated by the European Convention of the Protection of Vertebrate Animals used for Experimental and other Scientific Purposes. Global SphK1$^{-/-}$ mice were kindly provided by R. Proia (National Institutes of Health, Bethesda, USA). Global Mfsd2b$^{-/-}$ were purchased from the MMRRC at UC Davis. The Sgpl vav-cre mice were generated by crossing Sgpl flox/flox mice (A. Billich, Novartis) with Commnd10Tg(vav1-iCre)A2Kio/J vav Cre transgenic mice. 4-deoxypyridoxine (DOP) (Sigma, St. Louis, USA), was administrated via drinking water at 3 mg/l (0.5 mg per kg body weight per day) for three weeks. Simultaneously with the DOP, mice received a diet with low vitamin B6. Diet-induced obesity was induced by feeding a diet with 60% energy from fat (35% fat, 5.228 kcal/kg, Altromin Spezialfutter GmbH & Co. KG, Lage, Germany) for 8 or 12 weeks as indicated. Sex was not considered in study design because of similar S1P values in male and females. The mechanism of this study is independent of sex. All mice were kept at 12-hour light/dark cycle, ambient temperature 20–24 °C, 45–65% humidity, water provided ad libitum, euthanasia by isoflurane.

### RBC isolation and preparation
RBC were isolated from whole blood by centrifugation at 1500 × g at 4 °C for 10 min. After washing with PBS twice, cell counts were determined in a scil Vet abc Plus (Scil animal care company GmbH, Viernheim, Germany).

### S1P loading
Washed RBCs (100 × 10⁶ cells/ml) were incubated with 1 µM sphingosine (D-erythro, Enzo Life Sciences GmbH, Lörrach, Germany) 1 µM FTY720 (Cayman Chemical, Ann Arbor, USA) or 1 µM FTY720-P (Cayman Chemical, Ann Arbor, USA) at 37 °C for 30 min in Tyrode buffer (NaCl 113 mM, KCl 4.7 mM, KH$_2$PO$_4$ 0.6 mM, Na$_2$HPO$_4$ 0.6 mM, MgSO$_4$ 1.2 mM, NaHCO$_3$ 12 mM, KHCO$_3$ 10 mM, HEPES 10 mM, Taurine 30 mM, pH 7.0 at 37 °C) under continuously gentle shaking (Thermoshaker, Haep Labor Consult, Bovenden, Germany). After incubation, RBCs were washed (300 × g, 10 min) and RBC pallet was dissolved in 1 ml MeOH to determine S1P level of loaded and unloaded RBCs by LC-MS/MS measurements. EtOH served as solvent control for sphingosine and FTY720, DMSO as control for FTY720-P (1/ 1000). The PP2A inhibitor okadaic acid (Sigma-Aldrich, St. Louis, USA) was applied in 2 nM 15 min before sphingosine incubation.

### S1P efflux assay
With or without previous sphingosine loading, RBCs (100 Mio) were washed twice and then incubated with or without 1% BSA (fatty-acid free, Serva, Hamburg, Germany) in 1 ml of Tyrode buffer at 37 °C for 20 min with gentle shaking. Alternatively, RBCs were incubated with 800 pmol per ml of the specific S1P antibody Sphingomab LT1002 (Lpath Inc., San Diego, USA) or an isotype control IgG1, κ (Biolegend, San Diego, USA). RBCs where centrifuged at 300 x g for 10 min. Supernatants were

transferred in new tubes and stored at −80 °C prior folch extraction. The pellet was immediately taken up in 1 ml ice-cold MeOH, spiked with 10 µL internal standard (1 µM C$_{17}$ S1P in MeOH, Avanti Polar Lipids Inc.) and precipitated overnight at −80 °C. Samples were centrifuged at 21.000 × g for 5 min at 4 °C, supernatants were transferred to 1.5 ml glass vials and stored at −80 °C until LC-MS/MS measurement. Lipids from buffer were extracted by folch-extraction. Briefly 200 µl supernatant was thoroughly mixed with 200 µl aq. NaCl (1.0 M), 400 µl MeOH, 200 µL aq. HCl (6.0 M) and 800 µl CHCl$_3$ and centrifuged (21.000 × g for 3 min, r.t.). The layers were separated and the aqueous layer was extracted with CHCl3 (300 µL). All organic phases were combined and dried by centrifugation in vacuo (Martin Christ Gefriertrocknungsanlagen GmbH, Osterode, Germany) at 200 g at 50 °C for 1 h. Residues were dissolved in 100 µl MeOH, transferred into 300 µL LC glass vials and stored at −80 °C until LC-MS/MS measurement.

### Transfection and culturing of HEK293 cells
HEK293 cells overexpressing mouse Mfsd2b in vector ORF-MFSD2B (Origene) were cultured under 300 µg/ml G418 selection. HEK293 Mfsd2b and HEK293 control cells were stably transfected with murine Sphk1 (Origene) cloned into pCDNA4-zeo mammalian expression vector and transfected using FuGENE (Promega). After selection with 500 µg/ml Zeocin (Invitrogen, Thermo Fisher Scientific, Germany) for three weeks, single clones were expanded and Sphk1 expression confirmed by qRT-PCR.

### Glucose uptake
Glucose uptake was measured with a luciferase based 2-deoxyglucose assay (2DG) (Promega, Madison, USA). Washed RBC (100 × 10⁶ cells/ml) were incubated with sphingosine in a zero glucose media (Tyrode, pH 7,0). Subsequently, 5 Mio RBCs were transferred per well in a white 96 well Optiplate™ (PerkinElmer, Waltham, Germany). In the case of HEK293 cells, 20 000 cells/well were seeded in 96 well plate for 24 h. Medium was changed to glucose-free, phenol red free DMEM (Thermo Fisher Scientific, Waltham, USA) 1 h before the assay followed by incubation with 1 µM sphingosine at 37 °C for 1 h and washing once thereafter. Glucose uptake experiments were started by adding 1 mM 2DG with and without 1% BSA (fatty-acid free, Serva, Hamburg, Germany) at 37 °C for 20 min. 2DG-6-phosphate (2DG6P) served as standard. Luminescence was measured in a CLARIOstar Plus microplate reader (BMG LABTECH GmbH, Offenburg, Germany). For TPA experiments, washed human RBC's (100 × 10⁶) were incubated in 1 ml zero glucose and pre-incubated for 20 min with 1 µM sphingosine before a five-minute stimulation with TPA followed by 10 min incubation with 0.5 mM 2DG at 37 °C. Luminescence was measured in 5 Mio RBC.

### LC-MS/MS
Chromatographic separation was performed on a LCMS-8050 triple quadrupole mass spectrometer (Shimadzu Duisburg, Germany) with a Dual Ion Source and a Nexera X3 Front-End-System (Shimadzu Duisburg, Germany). Chromatographic separation for S1P were performed with a 2 × 60 mm MultoHigh 100 RP18-3 µm column (CS Chromatographie Service, Langerwehe, Germany) at 40 °C. Mobile phases consisted of [A] MeOH and [B] aq. HCO2H (1% v/v) and the following gradient settings were used: [A] increased from 10% to 100% over 3 min (B.curve = −2) and returned to 10% from 8.01 min to 10 min prior next injection. Flow rate was 0.4 ml/min and injection volume of all samples was 10 µl. MS settings were the following: Interface: ESI, nebulizing gas flow: 3 l/min, heating gas flow: 10 l/min, interface temperature: 300 °C, desolvation temperature: 526 °C, DL temperature: 250 °C, heat block temperature: 400 °C, drying gas flow: 10 l/min. Data were collected using multiple reaction monitoring (MRM) and positive ionization $[M + H]^+$ was used for qualitative analysis and quantification. The following MRM transitions were used for quantification: m/z = 380 → 264 or 82 for S1P (R$_t$ = 2.67 min) and m/z = 366 → 250 for C$_{17}$ S1P

($R_t$ = 2.55 min). Standard curves were generated by measuring increased amounts of analytes (10 nM to 50 μM S1P, Avanti Polar Lipids Inc.) with internal standard (100 nM $C_{17}$ S1P) in MeOH. Metabolome primary data were analysed and further processed with LabSolutions 5.99 (Shimadzu Deutschland GmbH, Duisburg, Germany) and further processed in Microsoft Excel. 1 pmol S1P/Mio RBC equals 21 μMol/L S1P calculated based on a MCV of 47.5 fl.

## Metabolic flux analysis by ion chromatography MS

Washed RBC (100 × $10^6$ cells/ml) with or without S1P loading were incubated with 6 mM of 1,2,3-$^{13}C_3$-D-glucose (Cambridge Isotope Laboratories, Inc., Tewksbury, USA) for the indicated times. After centrifugation (300 × $g$, 10 min) and washing, the RBC pellet was dissolved in 1 ml of ice cold MeOH/CHCl$_3$/H$_2$O (2:1:1) frozen and stored at −80 °C. For metabolite extraction 57 μl CHCl$_3$ (−20 °C) and 560 μl ice cold MS grade water containing 4.6 μM Adenosine-5′-O-1-Thiotriphosphate as internal standard were added to each sample to induce phase separation. After mixing and 10 min incubation on ice, the sample was centrifuged at 10.000 × $g$ for 2 min and the aqueous phase was collected Additional 460 μl pure water were added to the remaining organic phase, mixed and centrifuged at 10.000 × $g$. The aqueous phase was combined with the previous phase and 650 μl pure water were added to dilute the organic proportion below 15%. The sample was frozen at −80 °C and subsequently dried by lyophilisation. The $^{13}C$ enrichment in lactate and PRPP was measured with anion exchange chromatography (Dionex ICS-6000 HPIC, Thermo Scientific) coupled to mass spectrometry (Q Exactive Plus quadrupole-Orbitrap mass spectrometer (Thermo Scientific)) as described[49]. Anion exchange chromatography was conducted on a Dionex IonPac AS11-HC column (2 mm × 250 mm, 4 μm particle size, Thermo Scientific) equipped with a Dionex IonPac AG11-HC guard column (2 mm × 50 mm, 4 μm, Thermo Scientific) at 30 °C. The mobile phase was established using an eluent generator with a potassium hydroxide cartridge to produce a potassium hydroxide gradient. The column flow rate was set to 380 μl min$^{-1}$ with a starting KOH concentration of 5 mM for one minute. The concentration was increased to 85 mM within 35 min and held for 5 min. The concentration was immediately reduced to 5 mM and the system equilibrated for 10 min. Spray stability was achieved with a make up flow consisting of methanol with 10 mM acetic acid delivered with 150 μl min$^{-1}$ by an AXP Pump. The electro spray was achieved in the ESI source using the following parameters: sheath gas 30, auxiliary gas 15, sweep gas 0, spray voltage - 2.8 kV, capillary temperature 300 °C, S-Lens RF level 45, and auxiliary gas heater 380 °C. For the untargeted approach the mass spectrometer operated in a combination of full mass scan and a data-dependent Top5 MS2 (ddMS2) experiment. The full scan (60−800 m/z) was conducted with a resolution of 140.000 and an automatic gain control (AGC) target of $10^6$ ions with a maximum injection time of 500 ms. The Top5 ddMS2 experiment was carried out with a resolution of 17.500 and an AGC target of $10^5$ ions and a maximum IT of 50 ms. The stepped collision energy was used with the steps (15,25,35) to create an average of NCE 25. Data analysis was conducted using Compound Discoverer (version 3.1, Thermo Scientific) using the "untargeted Metabolomics workflow". Compound identification was achieved on the level of mass accuracy (MS1 level), fragment mass spectra matching (MS2 level) and retention time comparison with authentic standards. For the enrichment analysis with stable heavy isotopes, the „stable isotope labeling" standard workflow for was chosen with the default settings 5 ppm mass tolerance, 30 % intensity tolerance and 0.1 % intensity threshold for isotope pattern matching. Maximum exchange rate was 95%.

## PP2A activity

PP2A was measured using a PP2A Immunoprecipitation Phosphatase Assay Kit (Merck Millipore, Burlington, USA) after incubation of RBC with the indicated substances and C6 ceramide (Cayman Chemical,

Ann Arbor, USA) as positive control in TBS buffer (200 mM Tris, 1500 mM NaCl, pH 7.4) at 37 °C for 30 min and lysis in 200 μl cold buffer containing 20 mM imidazole, 2 mM EDTA, 2 mM EGTA, and total protease inhibitor cocktail (Thermo Fisher Scientific, Waltham, USA) according to the manufacturer's instructions. On some occasions, PP2A was first isolated from murine C57Bl6 RBCs by immunoprecipitation and then the beads incubated with substances. In the case of human recombinant PP2A catalytic subunit (Cayman Chemical, Ann Arbor, USA) 0.58 mU in 100 μl were incubated for 10 min with all substances. Phosphatase activity was determined by the enzymatic conversion of threonine phosphopeptide to anorganic phosphate and malachite green detection solution as measured by at 650 nm in a microplate reader (BMG LABTECH GmbH, Offenburg, Germany).

## GLUT1 flow cytometry

Surface GLUT1 localization in human RBC was measured using the receptor-binding domain of a recombinant envelope glycoprotein from human T lymphotrophic virus fused to EGFP (H2-EGFP) kindly provided by Dr. V. Petit, Metafora Biosystems, as described[50,51]. Briefly, RBCs were incubated with H2-EGFP in PBS containing 0.33 mg/ml BSA and 1 mM EDTA at 37 °C for 20 min. Afterwards, RBCs are washed twice with PBS and re-suspended in 400 μl FACS buffer (NaCL 140 mM, $H_2Na_2O_6P$ 13 mM, EDTA 1.075 mM, KCL 5.5 mM, $NaH_2PO_4 \cdot H_2O$ 1.6 mM, NaF 75.55 mM). GFP signal was detected by flow cytometry on a Gallios Flow Cytometer and analyzed by 1.2 Data Acquisition & Analysis Software (Beckman Coulter, Brea, USA).

## GLUT4 flow cytometry

Cell surface GLUT4 flow cytometry was performed with an antibody that recognizes an extracellular domain of the transporter (LM048) as published[22]. On site validation was performed with the L6-GLUT4myc rat myoblast cell line (Kerafast, Boston, USA) that overexpresses GLUT4. Briefly, 450.000 L6 cells were kept in phenol-free DMEM medium without glucose supplemented with 1% normal goat serum (NGS) for 3 h at 37 °C, 5% CO2 in a humidified atmosphere and incubated with LM048 for additional 20 min at 37 °C. Fixation in 3.5% final concentration of PFA was performed for 10 min on ice. After washing (400 × $g$, 5 min), L6 cells were incubated with a biotinylated goat anti-human IgG Fc secondary antibody (1:100, eBioscience, ThermoFisher Scientific, Waltham, USA) for 1 h at 4 °C in the same medium, followed by streptavidin PE (1:1000, BD Biosciences, Heidelberg, Germany) for 30 min at 4 °C. Negative controls were secondary antibody alone as well as anti-GLUT4 PE (1:200, clone IF8, Santa Cruz Biotechnology, Heidelberg, Germany) that recognizes an intracellular epitope of GLUT4. Intracellular staining was also performed with fixed/washed L6 cells after treatment. Cell permeabilization was achieved by incubation in 0.1% Triton/PBS for 5 min at RT. Cells were then blocked in PBS with 1 % BSA and FBS for 1 h at RT and stained. For GLUT4 staining on RBCs, ~40 × $10^6$ RBCs were washed once in 1 ml cold PBS and fixed in 1 ml 2% PFA for 24 h at room temperature. For GLUT4 staining on S1P-loaded RBCs 100 Mio RBCs were pre-incubated in 1 ml of Tyrode buffer at 37 °C for 10 min with gentle shaking, followed by an additional incubation with vehicle or 1 μM sphingosine for another 30 min at 37 °C. 10 × $10^6$ RBCs were fixed in 2 % final PFA for 24 h at room temperature, spun down (500 × $g$, 6 min) and resuspended in 1 ml PBS at RT. 2.5 × $10^6$ RBCs were washed in phenol-free DMEM medium without glucose supplemented with 1% NGS and stained with LM048 at 37 °C in the same medium for 20 min. After washing, RBCs were incubated with a biotinylated goat anti-human IgG Fc secondary antibody followed by streptavidin PE as described above. Fluorescence was detected by a Gallios Flow Cytometer (Beckman Coulter, Brea, USA).

## Western blotting

Human RBC pellet (10 × $10^6$) after the TPA experiment (see 'glucose uptake') was lysed in equal volume of NP-40 lysis buffer (1% IGEPAL,

25 mM HEPES, 120 mM NaCl, 1 mM EDTA, pH 7.4) and 4x Laemmli sample buffer was added for ten minutes at 50 °C. Samples were separated on a 4–20% Tris-gradient SDS-PAGE, transferred to a PVDF-membrane overnight and membranes probed for phospho-Serine 226-GLUT1 (ABN991; 1:200) and beta-actin (A1978; 1:100), both from Sigma-Aldrich, St. Louis, USA. Original blots are shown in supplementary Fig. 5.

## HbA1c
HbA1c levels were measured by Mouse Glycated Hemoglobin A1c (HbA1c) ELISA Kit (MyBioSource San Diego, USA). Human HbA1c was determined by clinical routine diagnostics.

## Glucose tolerance tests
Glucose tolerance tests (GTT) were performed in overnight fasted (16 h) mice. Blood was drawn from the tail vein and blood glucose concentration determined at 0, 15, 30, 60, 90, 120 and 240 min after intraperitoneal glucose injection (Sigma-Aldrich, St. Louis, USA; 2 g/kg body weight) using StatStrip Xpress®2 glucometer (Nova Biomedical, USA).

## Lipid peroxidation
Lipid peroxidation of RBCs was determined by extracting and measuring malondialdehydes (MDAs) from fresh RBC as described[52]. Briefly, RBC were mixed with TCA (28% w/v) in a 2:1 ratio, vortexed and incubated for 10 min at RT. After centrifugation (16,100 × g, 10 min, 18 °C), the MDA extracts (supernatants) were collected. Extracts were mixed in a 4:1 ratio with either TBA (1%, dissolved in 0.05 mol/L NaOH) or 0.05 mol/L NaOH as blank. MDA standards were prepared and treated as all samples. RBC extracts, sample blank, and MDA standards were boiled for 15 min and immediately cooled on ice for 10 min. 100 μl were transferred into a 96-well plate and TBARS quantified with a microplate reader (BMG LABTECH GmbH, Offenburg, Germany; MARS Data Analysis Software) at 453 and 532 nm from the MDA standard curve.

## Statistics
Statistical analyses were performed with GraphPad Prism 9.3.1. Individual tests are indicated in each figure legend.

## Reporting summary
Further information on research design is available in the Nature Portfolio Reporting Summary linked to this article.

## Data availability
The data supporting this study can be found in the figures and supplementary information. Source data are provided with this paper.

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

## Acknowledgements

We gratefully acknowledge excellent technical help by Kerstin Petsch and Rabia Taskesen. This work was funded by the German Research Foundation (GRK 2098, project 10; LE 940/7-1, PO 2247/2-1, and TRR259, project B10). We thank Dr. Vincent Petit for providing the HRBDEGFP protein. We thank smart.servier.com for medical graphics. LM608 was a kind gift from Joseph Rucker, Integral Molecular, Philadelphia, PA 19104. Metabolite analyses were supported by the CEPLAS Plant Metabolism and Metabolomics laboratory funded by the German Research Foundation under Germany´s Excellence Strategy – EXC-2048/1 – project ID 390686111.

## Author contributions

N.T. and N.H.S.: designed research, performed research, analysed data, and wrote the manuscript. P.W., K.v.W.L., A.W., M.K.N., J.D.-J., J.V., P.W., S.W., P.K.: analyzed data. S.G., M.H.G., L.D., T.B., M.K., A.P.: contributed vital new reagents or analytical tools. B.L.: designed research, analyzed and interpreted data, wrote the manuscript.

## Funding

## Competing interests

The authors declare no competing interests.
