## [Peer Review File · Nature Communications]

REVIEWER COMMENTS

Reviewer #1 (Remarks to the Author):

This manuscript contains extensive data which suggest that intracellular S1P of RBCs regulate PP2A phosphatase activity which then somehow impacts cell surface glucose transporter activity. The authors used pharmacological tools as well as some genetic models and biochemical assays to develop their hypothesis. They also suggest that the rate of glucose uptake by RBCs can impact plasma Hb1c levels in diabetes.

The concept is potentially novel and the manuscript is well written. This said, there are some major weaknesses in the data and the critical evidence needed to prove the model. In its current form, the data do not fully support the conclusions.

1. The major weakness is that the changes in glucose uptake (2DG) are modest and in some cases even though the authors claim statistical significance, I am not convinced that it is biologically meaningful. The rate of 2DG uptake seems quite low. The data should also be reviewed by a professional statistician to ensure it is sufficiently powered. Can the authors demonstrate biochemical evidence of Glucose transporter modifications by PP2a (either direct dephosphorylation, level on the cell surface or expression levels)? If somehow S1P is modulating PP2A and that is somehow influencing Glut1 and/or 4 activity and/or expression and/or localization, this should be directly shown by biochemical methods and cell biological approaches (i.e. confocal immunofluorescence microscopy).

2. The authors cite the work from Dr. Xia's group suggesting RBC S1P regulation of 2,3-BPG and HB/ O₂ interactions. They do not address whether this published mechanism is relevant to their studies. This is critical if we are to believe that RBC-derived S1P is playing an important cellular signaling function.

3. The inclusion of albumin concomitantly with extracellular sphingosine to demonstrate reduction of intracellular S1P is complicated by the fact that albumin will buffer sphingosine and therefore reduce the effective concentration of substrate that is uptaken by RBCs. In addition, extracellular fatty acid free albumin will extract sterols and other lipids and therefore could influence membrane lipid rafts and therefore have a non-specific effect on RBC physiology. It is perhaps better to use non-metabolizable sphingosine analogs - i.e. stereoisomers that are not acted on by the sphingosine kinase to see if the effects are still observed.

4. PP2A is regulated by sphingosine as shown by other publications listed below. Some of the authors' data on sphingoid bases contradict these studies. Any explanations?

Xie T, Chen C, Peng Z, Brown BC, Reisz JA, Xu P, Zhou Z, Song A, Zhang Y, Bogdanov MV, Kellems RE, D'Alessandro A, Zhang W, Xia Y. Erythrocyte Metabolic Reprogramming by Sphingosine 1-Phosphate in Chronic Kidney Disease and Therapies. *Circ Res*. 2020 Jul 17;127(3):360-375. doi: 10.1161/CIRCRESAHA.119.316298. Epub 2020 Apr 14. PMID: 32284030.

Saddoughi SA, Gencer S, Peterson YK, Ward KE, Mukhopadhyay A, Oaks J, Bielawski J, Szulc ZM, Thomas RJ, Selvam SP, Senkal CE, Garrett-Mayer E, De Palma RM, Fedarovich D, Liu A, Habib AA, Stahelin RV, Perrotti D, Ogretmen B. Sphingosine analogue drug FTY720 targets I2PP2A/SET and mediates lung tumour suppression via activation of PP2A-RIPK1-dependent necroptosis. *EMBO Mol Med*. 2013 Jan;5(1):105-21. doi: 10.1002/emmm.201201283. Epub 2012 Nov 25. PMID: 23180565; PMCID: PMC3569657.

Habrukowich C, Han DK, Le A, Rezaul K, Pan W, Ghosh M, Li Z, Dodge-Kafka K, Jiang X, Bittman R, Hla T. Sphingosine interaction with acidic leucine-rich nuclear phosphoprotein-32A (ANP32A) regulates PP2A activity and cyclooxygenase (COX)-2 expression in human endothelial cells. *J Biol Chem*. 2010 Aug 27;285(35):26825-26831. doi: 10.1074/jbc.M110.147058. Epub 2010 Jun 17. PMID: 20558741; PMCID: PMC2930681.

Camp SM, Marciniak A, Chiang ET, Garcia AN, Bittman R, Polt R, Perez RG, Dudek SM, Garcia JGN. Sphingosine-1-phosphate receptor-independent lung endothelial cell barrier disruption induced by FTY720 regioisomers. *Pulm Circ*. 2020 Feb 10;10(1):10.1177_2045894020905521. doi: 10.1177/2045894020905521. PMID: 32095229; PMCID: PMC7011338.

5. I am unclear from the authors' work how small changes in RBC glucose uptake is impacting big changes in the glycation of Hb in diabetics. This seems to be a major conclusion with important clinical and pathophysiological implications that are not fully supported by data.

6. Can the authors demonstrate changes in phosphorylation or cell surface localization of Glut4 in murine RBCs in some key genetic models ?

Reviewer #2 (Remarks to the Author):

Thomas and colleagues describe a role for Sphingosine 1 phosphate (S1P) in GLUT suppression via PP2A, a mechanism they report to participate in the regulation of glycemia in diabetic red blood cells. First of all, I have to disclose my positive bias towards mechanistic studies that focus on novel pathways relevant to red blood cell biology. Overall, this is a very interesting manuscript. However, a few additional experiments are recommended, especially to validate the findings on glucose uptake and reconcile the current findings with prior reports on increased glycolytic activity in RBCs exposed to hypoxia as a function of S1P levels (e.g., in Sphk1 mice, as referenced). In addition, while the phenomenon is documented by relatively solid evidence, it is unclear whether and to what extent the S1P-PP2A-GLUT1 axis has any functional effect relevant to disease (only evidence provided relates to Hb1Ac levels in a murine model of diabetes). A few detailed comments are provided below.

Major comments:

- 1) Previous studies (e.g., reference 19 in the reviewed version of the study) have clearly documented a mechanistic linkage between S1P levels and glycolytic fluxes in the mature RBC, especially in response to hypoxia. How do the authors reconcile their findings with these previous studies? I would recommend performing experiments with stable isotope-labeled tracers (U-13C-glucose) to validate their findings on glucose uptake and metabolism.
- 2) In the introduction section, when referring to glycolysis generating NAD for ferric iron reduction (via methemoglobin reductase, I assume), the authors should specify that they are referring to the reduced form of NAD (NADH). Other NAD(P)H diaphorases exist in the mature RBC. Metabolic switch from glycolysis to the PPP (which has been shown to be modulated by S1P levels – Sun et al. Nat Comm 2016) may play a role in the mechanism the authors are trying to suggest to be at play in this study.
- 3) Results section: which S1P concentration was used in the experiments involving exogenous supplementation? S1P reaches critical micellar concentration at 12 micromolar. Please, convert the units pmol/million RBCs to a more standardized concentration throughout the manuscript.
- 4) Given the relevance to the conclusion that S1P affects GLUT-dependent glucose uptake, measurements of proton efflux rates are insufficient and indirect indicators of glycolytic rates. As such, the Seahorse data have to be complemented by steady state and 13C glucose tracing experiments. I think this is the most critical missing piece of evidence in this otherwise really interesting story.
- 5) Where the Sphk1 $-/-$ mice missing Sphk1 just in the erythroid compartment (e.g., EpoR-Cre Sphk1 f/f) or constitutively missing it in all tissues?
- 6) Previous studies (already referenced here) have shown that S1P can stimulate PKA activity. Do the authors think that the observed effect on PP2A is direct or rather involving intermediate kinase mediators?
- 7) I have some issue with statements like “S1P-loading decreased GLUT1 expression on the RBC surface by ~50%”, given the fact that mature RBCs lack de novo protein synthesis capacity. As such, are the

authors proposing that (i) S1P affects hematopoietic stem cell maturation by decreasing GLUT1 expression during the late stages of erythroid differentiation? (ii) that S1P modulates untimely removal (degradation?) of GLUT1, in like fashion to prior studies on adenosine triggering the phosphorylation and proteasomal degradation of its own ENT1 transporter (Song et al. Nat Commun. 2017 Feb 7;8:14108.)? In any case, a clear mechanism is missing.

8) Are the diabetic mice (DIO model) also showing dysfunctional oxygen kinetics that promote tissue hypoxia (e.g., hypoxyprobe staining of high oxygen consuming organs such as brain, heart, kidneys)? These experiments are relevant to understand whether the observed phenomenon (S1P regulates glucose uptake in the erythrocyte) has any functional consequences on the RBC capacity to carry and deliver oxygen to peripheral tissues.

9) The studies on Mfsd2b ^{-/-} and DIO diet are really elegant. Congratulations!

10) Discussion section: "This is, to our knowledge, the first report that changes in intracellular S1P concentrations dynamically regulate glucose metabolism in RBC." Technically, this is not true. The authors reference Sun et al. Nat Comm 2016, by the title "Sphingosine-1-phosphate promotes erythrocyte glycolysis and oxygen release for adaptation to high-altitude hypoxia." Please, specify "in the context of diabetes".

11) In the discussion section the authors suggest that S1P role in regulating glucose influx would contribute to hyperglycemia-induced lipid peroxidation, loss of glutathione S-transferase or reductase activities, hemolysis and eryptosis. Experiments aimed at assessing at least hemolysis and lipid peroxidation, glutathione levels and reactive oxygen species should be performed.

Minor

1) Some references are repeated, while other critical ones are missing. For example, reference 19 and 30 refer to the same paper, while other relevant references from Dr Xia's group are missing, such as:

a. Sun et al. Sci Rep. 2017 Nov 10;7(1):15281.

b. Xie et al. Circ Res. 2020 Jul 17;127(3):360-375.

c. Zhang et al. J Clin Invest . 2014 Jun;124(6):2750-61.

d. Zhao et al. FASEB J . 2018 May;32(5):2855-2865.

These manuscripts should be discussed in light of the role of S1P in chronic kidney disease (the linkage between of diabetes and CKD is well-documented) and sickle cell disease.

Reviewer #3 (Remarks to the Author):

The authors claimed that glucose uptake in erythrocytes is negatively modulated by intracellular S1P contents via PP2A activities.

The findings are interesting, however, they cannot provide enough evidence for the proposed mechanisms. I suspect the presence of some indirect mechanisms. Moreover, they could not show any physiological significance of their findings. As they have shown in Figure 7A, the intracellular S1P levels had no impact on glucose metabolism.

The specific points are follows.

Major points

#1. PP2A can be modulated by ceramides and since the metabolism of ceramides and S1P is close, I think that the indirect pharmacological or genetic modulations of S1P cannot provide enough evidence for the modulation of GLUTs by intracellular S1P. The authors should measure ceramide levels and should investigate the modulation of glucose uptake by S1P itself. Of course, I understand that extracellularly administration of S1P can be hardly transported through cell membrane, from my experience, S1P can be somehow transported across cell membrane. Moreover, caged S1P can be moved trans-membranously.

#2. The authors should also show the dependency of the modulation of glucose uptake on intracellular concentration of S1P in pharmacological modulations of intracellular S1P and/or erythrocytes from heterogeneous knockout of Mfsd2b.

#3. I have concerns that the extents of the modulations of intracellular S1P on those of the influences on glucose uptake are not concordant.

Fig 1A, C: As much as 15 fold increase of S1P resulted in 34 % decrease of glucose uptake.

Fig 1B, C: Only 40% decrease of S1P restored 34 % decrease of glucose uptake.

Fig 2D, E; Only 1.8 fold increase of S1P resulted in 26 % decrease of glucose uptake.

Fig 2G, I: 13 fold decrease of S1P resulted in 34 % increase in glucose uptake.

Fig 3A, C: 40-fold increase of S1P resulted in 20 % decrease in glucose uptake.

I have concerns that these data suggested that the modulations of S1P might not directly affect glucose uptake, but some indirect effects such as physical modulations of membranes by extremely modulated S1P.

#4, Fig. 3D, 3F. If the mechanisms for the regulation of glucose uptake by S1P are not limited to erythrocytes, how about other organs involved in glucose metabolism, such as liver, muscle, and adipose tissues?

#5. Fig 4E. What vehicles are used to deliver S1P or other lipids ? S1P might not be resolved in solution, such as H₂O. The authors present these experiments for the evidence of S1P involvement of PP2A activity, however, these experiments are not physiological.

#6 . Please explain the meaning of Fig. 7 in the context of the manuscript in more detail. Especially the sentence of “This clearly demonstrated that high intracellular S1P protected RBC from pathological hemoglobin glycation in chronic hyperglycemia.”

#7. I think that Fig 7A suggest that the phenomena revealed by the authors had little impact on physiological glucose metabolism.

What is the physiological significance of this finding?

#8. Characteristics of the subjects in Figure 6 should be provided. Why intracellular levels of S1P are modulated in the subjects of type 2 diabetes and DIO mice?

Are levels of Mfsd2B different?

#9. The modulations of glut1/4 mediated by intracellular S1P/PP2A pathway might influence the ability of RBC. I think this might have physiological significance, not glucose metabolism.

Were the shape, erythrocyte-related laboratory parameters such as MCV, MPV, and Mdw, vulnerability of erythrocytes, and ability to carry oxygen in the erythrocytes modulated when intracellular S1P was modulated?

Was hemolysis observed in Mfsd2b KO mice reared HFD or in extremely high glucose conditions caused by the injection of streptozotocin, in which the physiological regulation of S1P-Glut axis was disturbed, as the authors described in the Discussion?

REVIEWER COMMENTS

Reviewer #1 (Remarks to the Author):

This manuscript contains extensive data which suggest that intracellular S1P of RBCs regulate PP2A phosphatase activity which then somehow impacts cell surface glucose transporter activity. The authors used pharmacological tools as well as some genetic models and biochemical assays to develop their hypothesis. They also suggest that the rate of glucose uptake by RBCs can impact plasma Hb1c levels in diabetes.

The concept is potentially novel and the manuscript is well written. This said, there are some major weaknesses in the data and the critical evidence needed to prove the model. In its current form, the data do not fully support the conclusions.

Thank you for the kind consideration of our data. Two major experiments have now provided crucial insight on mechanisms involved and their physiological relevance:

1) We have addressed the consequences of reduced glucose uptake by monitoring metabolic fluxes with $C^{13}C_3$ (1,2,3) glucose. Indeed, these experiments showed dramatic S1P effects on glucose metabolism that were reflected in a 50% increase in $C^{13}C_3$ lactate/ $C^{13}C_2$ lactate ratio representative of enhanced glycolysis and the 400% reduction in $C^{13}C_2$ phosphoribosyl pyrophosphate relative exchange rate representative of reduced flux through the pentose phosphate pathway.

2) We have obtained a worldwide unique antibody that recognises only an EXTRACELLULAR epitope of GLUT4 enabling us to monitor the dynamics of GLUT4 presence of on the cell surface under treatment and in genetic S1P models.

We provide the results below and in the manuscript. Thank you for having us perform these experiments as they have greatly increased impact and significance of the study.

1. The major weakness is that the changes in glucose uptake (2DG) are modest and in some cases even though the authors claim statistical significance, I am not convinced that it is biologically meaningful. The rate of 2DG uptake seems quite low. The data should also be reviewed by a professional statistician to ensure it is sufficiently powered.

Glucose uptake measurements based on 2DG in erythrocytes are not trivial and yield differences in absolute values. Nevertheless, in the figure below we have plotted the same 2DG experiments performed again and again with RBC preparations from all “wild type” mice of this study irrespective of genetic background performed over 2 years (n=35; Data are presented as mean±sd and tested with paired one-way ANOVA). We see a highly significant and sufficiently powered reduction of glucose uptake by S1P loading and its significant but not complete restoration by unloading.

To support this and follow up on the metabolic consequences for RBC physiology, we monitored glucose metabolism using the $C^{13}C_3$ (1,2,3) D-glucose isotope. There we observed large differences in the metabolic flux in glycolysis and PPP as detailed above and under point 5. In brief, there is a clear increase in glycolysis and a decrease in PPP flux very similar in nature to that in the cited work from Dr. Xia's group.

Can the authors demonstrate biochemical evidence of Glucose transporter modifications by PP2a (either direct dephosphorylation, level on the cell surface or expression levels)? If somehow S1P is modulating PP2A and that is somehow influencing Glut1 and/or 4 activity and/or expression and/or localization, this should be directly shown by biochemical methods and cell biological approaches (i.e. confocal immunofluorescence microscopy).

The most convincing method on such occasions is flow cytometry for the transporter in question. However, it is also the most challenging one as there is no antibody worldwide but one that recognizes EXTRACELLULAR GLUT4. We were lucky to obtain it this occasion: it is LM048 from Joe Rucker's company (Tucker DF, Sullivan JT, Mattia KA, Fisher CR, Barnes T, Mabila MN, Wilf R, Sulli C, Pitts M, Payne RJ, Hall M, Huston-Paterson D, Deng X, Davidson E, Willis SH, Doranz BJ, Chambers R, Rucker JB. Isolation of state-dependent monoclonal antibodies against the 12-transmembrane domain glucose transporter 4 using virus-like particles. *Proc Natl Acad Sci U S A*. 2018 May 29;115(22):E4990-E4999. doi: 10.1073/pnas.1716788115. Epub 2018 May 16. PMID: 29769329; PMCID: PMC5984492).

Using this antibody, we have now shown how S1P affects GLUT4 cell surface localization (Fig. 1G), its dependence on PP2A activity (Fig. 5C) and its reduction on RBC from MFSD2B-deficient mice (Fig. 3C inset).

2. The authors cite the work from Dr. Xia's group suggesting RBC S1P regulation of 2,3-BPG and HB/O2 interactions. They do not address whether this published mechanism is relevant

to their studies. This is critical if we are to believe that RBC-derived S1P is playing an important cellular signaling function.

Indeed, our new experiments show a similar increase in glycolysis and a decrease in PPP very similar to the cited work from Dr. Xia's group (see above). We have also measured 2,3-BPG as requested and find it increased with higher RBC S1P after sphingosine treatment as well as in *Mfsd2b*^{-/-} RBC (see below). Data are now provided in supplemental figure 3.

Supplemental figure 3: A) 2,3-BPG relative relative exchange rate (time kinetic) in RBC with C¹³C₃ glucose with and without sphingosine preincubation. B) Same C¹³C₃ glucose experiment (20 min) with *Mfsd2b* wild type and ko mice without sphingosine. Data are presented as mean±sd and tested with two-way ANOVA (A) and Mann-Whitney U test (B).

3. The inclusion of albumin concomitantly with extracellular sphingosine to demonstrate reduction of intracellular S1P is complicated by the fact that albumin will buffer sphingosine and therefore reduce the effective concentration of substrate that is uptaken by RBCs. In addition, extracellular fatty acid free albumin will extract sterols and other lipids and therefore could influence membrane lipid rafts and therefore have a non-specific effect on RBC physiology.

We have obviously been too brief on the method generating some confusion: No buffering of sphingosine by albumin can take place in our experiments as RBCs are first incubated with sphingosine for a defined period of time, then washed twice to eliminate any residual sphingosine and only thereafter incubated with albumin. We have now specified this under the methods section "S1P efflux assay". Also, non-specific effects were excluded by experiments with albumin alone (no loading) throughout Fig. 1 A-C and by showing specificity of the effect by extracting specifically S1P by the S1P-neutralizing antibody Sphingomab in Fig. 1E, F (used for this purpose in ref 21 and in Ader, I., C. Gstalder, P. Bouquerel, M. Golzio, G. Andrieu, S. Zalvidea, S. Richard, R. A. Sabbadini, B. Malavaud and O. Cuvillier (2015). "Neutralizing S1P inhibits intratumoral hypoxia, induces vascular remodelling and sensitizes to chemotherapy in prostate cancer." *Oncotarget* 6(15): 13803-13821).

It is perhaps better to use non-metabolizable sphingosine analogs - i.e. stereoisomers that are not acted on by the sphingosine kinase to see if the effects are still observed.

We have gladly followed up on the suggestion of non-metabolizable sphingosine analogues and performed experiments with FTY720 and FTY720-P (now to be found as new Fig. 4). The

charm of this approach is that FTY720 is phosphorylated only by sphingosine kinase 2 that is not expressed in RBCs. This is confirmed by the complete lack of conversion of FTY720 to FTY720-P in Sphk2ko RBC. Accordingly, in Sphk2ko RBC there is no reduction of glucose uptake by FTY720 but a very well with FTY720-P. Finally, in favour of S1P rather than sphingosine being the biologically active determinant of glucose uptake is also the observation that in SphK1-/- RBC there is no reduction of glucose uptake despite considerable sphingosine accumulation in the absence of S1P generation (Fig. 2I, J, K).

4. PP2A is regulated by sphingosine as shown by other publications listed below. Some of the authors' data on sphingoid bases contradict these studies. Any explanations?

We appreciate the opportunity to clarify this particularly as we have now added data on FTY720 that is widely accepted as a PP2A activator.

Here are our thoughts on the 4 papers the reviewer cites. In brief, there are apparently 2 mechanisms: One involves binding of certain sphingoid bases and analogs (sphingosine, FTY720 and DMS but not FTY720-P or S1P) to the PP2A inhibitors SET and ANP32A which relieves PP2A from inhibition hence activating it. The second one (the one we describe) is the direct of the catalytic subunit of PP2A by S1P and FTY720-P (but not sphingosine or FTY720) in the absence of SET and ANP32A. This has gone into the discussion section of the manuscript.

Here are the details:

Xie T, Chen C, Peng Z, Brown BC, Reisz JA, Xu P, Zhou Z, Song A, Zhang Y, Bogdanov MV, Kellems RE, D'Alessandro A, Zhang W, Xia Y. Erythrocyte Metabolic Reprogramming by Sphingosine 1-Phosphate in Chronic Kidney Disease and Therapies. *Circ Res.* 2020 Jul 17;127(3):360-375. doi: 10.1161/CIRCRESAHA.119.316298. Epub 2020 Apr 14. PMID: 32284030.

Here, the authors suggest that S1P inhibits PP2A when they exogenously add 6 μ M S1P to RBC. The major problem is that 6 μ M S1P clearly induces haemolysis (see our suppl. Fig. 2A). This would, of course, reduce the amount of PP2A in the authors' subsequent antibody-based PP2A capture assay (96-well plate to which the RBC lysate is added). Hence the authors would have observed pseudo inhibition due to less PP2A being present.

Also, Xie's study has performed no experiments on PP2A in any genetic or pharmacological models except an erythrocyte-specific Sphk1ko where PP2A was apparently unchanged which contradicts the inhibition finding in the same paper.

Saddoughi SA, Gencer S, Peterson YK, Ward KE, Mukhopadhyay A, Oaks J, Bielawski J, Szulc ZM, Thomas RJ, Selvam SP, Senkal CE, Garrett-Mayer E, De Palma RM, Fedarovich D, Liu A, Habib AA, Stahelin RV, Perrotti D, Ogretmen B. Sphingosine analogue drug FTY720 targets I2PP2A/SET and mediates lung tumour suppression via activation of PP2A-RIPK1-dependent necroptosis. *EMBO Mol Med.* 2013 Jan;5(1):105-21. doi: 10.1002/emmm.201201283. Epub 2012 Nov 25. PMID: 23180565; PMCID: PMC3569657.

Here, the authors convincingly show that FTY720 displaces/competes with SET for the PP2A holoenzyme resulting in reversal of inhibition and thus activation. Our data indicate another, different mechanism where S1P (and C6 ceramides) directly binds to and activates the PP2A catalytic subunit in the absence of SET.

Habrukowich C, Han DK, Le A, Rezaul K, Pan W, Ghosh M, Li Z, Dodge-Kafka K, Jiang X, Bittman R, Hla T. Sphingosine interaction with acidic leucine-rich nuclear phosphoprotein-32A (ANP32A) regulates PP2A activity and cyclooxygenase (COX)-2 expression in human endothelial cells. *J Biol Chem.* 2010 Aug 27;285(35):26825-26831. doi: 10.1074/jbc.M110.147058. Epub 2010 Jun 17. PMID: 20558741; PMCID: PMC2930681

Here, Tim Hla's group shows that another PP2A inhibitor (ANP32A) binds to biotinylated sphingosine but not to S1P. In addition, cells or pure ANP32A were treated with N,N-Dimethylsphingosine (although neither sphingosine nor S1P were tested) and PP2A activity was found to be increased. The authors conclude that PP2A is relieved from inhibition by ANP32A hence leading to its activation. This is very similar to the study on SET above. In contrast, we find S1P (but not sphingosine) to directly bind and activate PP2A in the absence of ANP32A.

Camp SM, Marciniak A, Chiang ET, Garcia AN, Bittman R, Polt R, Perez RG, Dudek SM, Garcia JGN. Sphingosine-1-phosphate receptor-independent lung endothelial cell barrier disruption induced by FTY720 regioisomers. *Pulm Circ.* 2020 Feb 10;10(1):10.1177_2045894020905521. doi: 10.1177/2045894020905521. PMID: 32095229; PMCID: PMC7011338.

Here, the authors show activation of PP2A by FTY720-treated endothelial cells by an unknown mechanism. This either of both mechanisms could be responsible – indirect SET- and/or ANP32A-based “PP2A relief” by FTY720 and the direct PP2A activation by FTY720-P that can be rapidly generated by HUVEC known to express Sphk1 and 2.

5. I am unclear from the authors' work how small changes in RBC glucose uptake is impacting big changes in the glycation of Hb in diabetics. This seems to be a major conclusion with important clinical and pathophysiological implications that are not fully supported by data.

This is an extremely important point. While 2-DG uptake is a surrogate for glucose import by GLUTs, it is useful only at on limited short term basis as it cannot be further metabolized after phosphorylation by hexokinase. Hence “small changes” in 2-DG uptake may underestimate “large” changes in genuine glucose import where the glucose is rapidly metabolized. In fact, we see this exactly to be the case. By monitoring metabolic fluxes with $C^{13}C_3$ glucose we have observed a much more dramatic S1P effect on glucose metabolism as reflected in the 50% increase in $C^{13}C_3$ lactate/ $C^{13}C_2$ lactate ratio and a 4-fold reduction of the $C^{13}C_2$ PRPP relative exchange rate (new Figure 1H, I). These data are now included in the results section and discussed.

6. Can the authors demonstrate changes in phosphorylation or cell surface localization of Glut4 in murine RBCs in some key genetic models?

Yes, we have done this using the extracellular epitope-specific GLUT4 antibody LM608 (please see answer to comment 1) and show reduction of GLUT4 on the surface of MFSD2B-deficient RBC (Fig. 3C inset).

In summary, we thank the reviewer for having us do the GLUT4 localization studies and the assessment of metabolic fluxes with $C^{13}C_3$ glucose. This has very much substantiated the physiological relevance of our study.

Reviewer #2 (Remarks to the Author):

Thomas and colleagues describe a role for Sphingosine 1 phosphate (S1P) in GLUT suppression via PP2A, a mechanism they report to participate in the regulation of glycemia in diabetic red blood cells. First of all, I have to disclose my positive bias towards mechanistic studies that focus on novel pathways relevant to red blood cell biology. Overall, this is a very interesting manuscript. However, a few additional experiments are recommended, especially to validate the findings on glucose uptake and reconcile the current findings with prior reports on increased glycolytic activity in RBCs exposed to hypoxia as a function of S1P levels (e.g., in Sphk1 mice, as referenced). In addition, while the phenomenon is documented by relatively solid evidence, it is unclear whether and to what extent the S1P-PP2A-GLUT1 axis has any functional effect relevant to disease (only evidence provided relates to Hb1Ac levels in a murine model of diabetes). A few detailed comments are provided below.

Thank you for the kind review of our work. As requested, we have performed metabolic flux studies with $C^{13}C_3$ (1,2,3) glucose to analyze glycolysis and pentose phosphate pathway fluxes and are happy to provide the data below and in the manuscript. Also, we have followed up on the question on pathophysiological consequences of our findings in respect to RBC biology in diabetes by assessing the effects on lipid peroxidation. The highly welcome results are presented below and in the manuscript. Thank you for having us perform these two series of experiments.

Major comments:

1) Previous studies (e.g., reference 19 in the reviewed version of the study) have clearly documented a mechanistic linkage between S1P levels and glycolytic fluxes in the mature RBC, especially in response to hypoxia. How do the authors reconcile their findings with these previous studies? I would recommend performing experiments with stable isotope-labeled tracers (U- ^{13}C -glucose) to validate their findings on glucose uptake and metabolism.

As recommended, we have addressed the metabolic consequences of reduced glucose uptake by monitoring metabolic fluxes with $C^{13}C_3$ (1,2,3) glucose. Indeed, these experiments showed dramatic S1P effects on glucose metabolism: we observed a 50% increase in $C^{13}C_3$ lactate/ $C^{13}C_2$ lactate ratio representative of enhanced glycolysis and a 400% reduction in $C^{13}C_2$ phosphoribosyl pyrophosphate relative exchange rate representative of reduced flux through the pentose phosphate pathway (new Figure 1H, I). These observations are very similar to the cited work from Dr. Xia's group cited by us and the reviewer.

2) In the introduction section, when referring to glycolysis generating NAD for ferric iron reduction (via methemoglobin reductase, I assume), the authors should specify that they are referring to the reduced form of NAD (NADH). Other NAD(P)H diaphorases exist in the mature RBC. Metabolic switch from glycolysis to the PPP (which has been shown to be modulated by S1P levels – Sun et al. Nat Comm 2016) may play a role in the mechanism the authors are trying to suggest to be at play in this study.

We have corrected accordingly and elaborate on the importance of our findings for lipid peroxidation in our answer to point 11.

3) Results section: which S1P concentration was used in the experiments involving exogenous supplementation?

We attempted S1P loading by direct supplementation of S1P instead of the detour through sphingosine. However, this was 5-fold less efficient (supplemental Fig. 2A), did not alter glucose uptake (supplemental Fig. 2B) and led to hemolysis from $> 4\mu\text{M}$ on (supplemental Fig. 2C). As sphingosine was much more efficient and did not lead to hemolysis (supplemental Fig. 2C) we continued using it as a tool to efficiently raise intracellular S1P.

S1P reaches critical micellar concentration at 12 micromolar. Please, convert the units pmol/million RBCs to a more standardized concentration throughout the manuscript.

1 pmol S1P/Mio RBC equals 21 $\mu\text{Mol/L}$ S1P calculated based on a MCV of 47.5 fl. Accordingly, S1P concentration in regular RBC is 2.98 μM as e.g. in Fig. 1a and reaches 104 $\mu\text{Mol/L}$ in *Mfsd2b* ko RBC (Fig. 3A). Of course, this is assuming that the RBC are mere volume units instead of a cell. There, S1P would be integrated in membranes and associated with proteins. We have provided this calculation both in the methods and in the figure legends.

4) Given the relevance to the conclusion that S1P affects GLUT-dependent glucose uptake, measurements of proton efflux rates are insufficient and indirect indicators of glycolytic rates. As such, the Seahorse data have to be complemented by steady state and ^{13}C glucose tracing experiments. I think this is the most critical missing piece of evidence in this otherwise really interesting story.

We completely agree and have performed the ^{13}C glucose tracing experiments. This has very much substantiated the physiological relevance of our study.

5) Where the *Sphk1* $^{-/-}$ mice missing *Sphk1* just in the erythroid compartment (e.g., *EpoR-Cre Sphk1 f/f*) or constitutively missing it in all tissues?

The *Sphk1* $^{-/-}$ mice are a global *Sphk1* knockout. This is now detailed in the methods.

6) Previous studies (already referenced here) have shown that S1P can stimulate PKA activity. Do the authors think that the observed effect on PP2A is direct or rather involving intermediate kinase mediators?

Our data argue for a direct effect because of the *in vitro* enzymatic studies where only the catalytic subunit and S1P were let to interact leading to increased PP2A activity. *In vivo*, of course, there may be other mechanisms as well. We have referred to this in the discussion.

7) I have some issue with statements like “S1P-loading decreased GLUT1 expression on the RBC surface by $\sim 50\%$ ”, given the fact that mature RBCs lack *de novo* protein synthesis capacity.

This is correct: we use now “localization” as a better term.

As such, are the authors proposing that (i) S1P affects hematopoietic stem cell maturation by decreasing GLUT1 expression during the late stages of erythroid differentiation? (ii) that S1P

modulates untimely removal (degradation?) of GLUT1, in like fashion to prior studies on adenosine triggering the phosphorylation and proteasomal degradation of its own ENT1 transporter (Song et al. Nat Commun. 2017 Feb 7;8:14108.)? In any case, a clear mechanism is missing.

We believe it is shuttling of GLUTs and possibly other transporters that is affected similar to the mechanism in tumour cells where synthetic sphingolipids such as FTY720 and other analogues have been shown to trigger the internalization of nutrient transporters by activating PP2A, which is then followed by mislocalization of PI(3,5)P2 generation by the lipid kinase PIKfyve and the triggering of cytosolic vacuolization by lysosomal fusion blockade (ref 27). Both pathways depend on PP2A activation by sphingolipids and exist side-by-side as the non-physiological short-chain C2 ceramide activates only the PP2A activity subset responsible for nutrient transporter loss but not for vacuolization, which may be due to different PP2A subsets being involved (ref 27, 28). We have discussed this in depth.

8) Are the diabetic mice (DIO model) also showing dysfunctional oxygen kinetics that promote tissue hypoxia (e.g., hypoxyprobe staining of high oxygen consuming organs such as brain, heart, kidneys)? These experiments are relevant to understand whether the observed phenomenon (S1P regulates glucose uptake in the erythrocyte) has any functional consequences on the RBC capacity to carry and deliver oxygen to peripheral tissues.

We have measured 2,3-BPG and find it increased with higher RBC S1P after sphingosine treatment as well as in *Mfsd2b*^{-/-} RBC (see below). Data are now provided in supplemental figure 3. As to pathophysiological role in diabetes, we have taken the suggestion under point 11 and have measured lipid peroxidation with quite interesting results.

Supplemental figure 3: A) 2,3-BPG relative relative exchange rate (time kinetic) in RBC with ¹³C₃ glucose with and without sphingosine preincubation. B) Same ¹³C₃ glucose experiment (20 min) with *Mfsd2b* wild type and ko mice without sphingosine. Data are presented as mean±sd and tested with two-way ANOVA (A) and Mann-Whitney U test (B).

9) The studies on *Mfsd2b*^{-/-} and DIO diet are really elegant. Congratulations!

Thank you.

10) Discussion section: "This is, to our knowledge, the first report that changes in intracellular S1P concentrations dynamically regulate glucose metabolism in RBC." Technically, this is not true. The authors reference Sun et al. Nat Comm 2016, by the title

“Sphingosine-1-phosphate promotes erythrocyte glycolysis and oxygen release for adaptation to high-altitude hypoxia.” Please, specify “in the context of diabetes”.

We have corrected accordingly.

11) In the discussion section the authors suggest that S1P role in regulating glucose influx would contribute to hyperglycemia-induced lipid peroxidation, loss of glutathione S-transferase or reductase activities, hemolysis and eryptosis. Experiments aimed at assessing at least hemolysis and lipid peroxidation, glutathione levels and reactive oxygen species should be performed.

This has been an excellent suggestion: the results of the experiments were delightful. As might have been expected, DIO lead to an increase in lipid peroxidation in RBC as assessed by measuring thiobarbituric acid reactive substances (TBARS). However, RBC from DIO-exposed *mfsd2b* ko mice harboring high S1P levels (and low HbA1c) had considerably less lipid peroxidation (new figure 8C). This suggests that their RBC are better protected against the adverse effects of diabetes associated with lipid peroxidation.

Minor

1) Some references are repeated, while other critical ones are missing. For example, reference 19 and 30 refer to the same paper,

This has been corrected.

while other relevant references from Dr Xia’s group are missing, such as:

- a. Sun et al. Sci Rep. 2017 Nov 10;7(1):15281.
- b. Xie et al. Circ Res. 2020 Jul 17;127(3):360-375.
- c. Zhang et al. J Clin Invest . 2014 Jun;124(6):2750-61.
- d. Zhao et al. FASEB J . 2018 May;32(5):2855-2865.

These manuscripts should be discussed in light of the role of S1P in chronic kidney disease (the linkage between of diabetes and CKD is well-documented) and sickle cell disease.

We have done this explicitly in the discussion section.

Reviewer #3 (Remarks to the Author):

The authors claimed that glucose uptake in erythrocytes is negatively modulated by intracellular S1P contents via PP2A activities.

The findings are interesting, however, they cannot provide enough evidence for the proposed mechanisms. I suspect the presence of some indirect mechanisms. Moreover, they could not show any physiological significance of their findings. As they have shown in Figure 7A, the intracellular S1P levels had no impact on glucose metabolism.

Thank you for your valuable comments and suggestions. We have diligently addressed them.

Although not asked by this reviewer (but by reviewers 1 and 2) we have addressed the physiological consequences of reduced glucose uptake in RBC by monitoring metabolic fluxes with $C^{13}C_3$ (1,2,3) glucose and observed a 50% increase in $C^{13}C_3$ lactate/ $C^{13}C_2$ lactate ratio representative of enhanced glycolysis and a 400% reduction in $C^{13}C_2$ phosphoribosyl pyrophosphate relative exchange rate representative of reduced flux through the pentose phosphate pathway (new Figure 1H, I).

The reviewer is also completely correct in that altered RBC S1P has no impact on whole body glucose metabolism (at least that monitored by GTT). However, we now provide very relevant data of this mechanism's beneficial impact on RBC lipid peroxidation in diabetes (answer to question #7).

The specific points are follows.

Major points

#1. PP2A can be modulated by ceramides and since the metabolism of ceramides and S1P is close, I think that the indirect pharmacological or genetic modulations of S1P cannot provide enough evidence for the modulation of GLUTs by intracellular S1P. The authors should measure ceramide levels and should investigate the modulation of glucose uptake by S1P itself. Of course, I understand that extracellularly administration of S1P can be hardly transported through cell membrane, from my experience, S1P can be somehow transported across cell membrane. Moreover, caged S1P can be moved trans-membranously.

These are excellent suggestions, indeed. We have now measured ceramides in the identical samples from which the sphingosine and S1P data in the manuscript stem from. Compared to S1P values, ceramides are ~25-fold lower and not influenced by sphingosine in our setting. Alas, the S1P issue is more difficult. We had attempted S1P loading by direct supplementation of S1P instead of the detour we have been taking through sphingosine. However, the same S1P concentration (1 μ M) that we used for sphingosine was 5-fold less efficient than sphingosine in increasing RBC S1P (supplemental Fig. 2A). This amount did not alter glucose uptake (supplemental Fig. 2B). We did increase the concentration but S1P from > 4 μ M on led to hemolysis (supplemental Fig. 2C). Sphingosine was thus much more efficient and did not lead to hemolysis which we certainly made sure of (supplemental Fig. 2C). Thus we continued using it as a tool to efficiently raise intracellular S1P particularly as we have done this in the past e.g. in experiments where we loaded S1P onto HDL by using

the efficient S1P production from sphingosine by RBC (ref). Also other groups have established this method years ago, so we just kept it unaltered.

We have taken the reviewer's thought very diligently into consideration and offer two other alternative experiments. Firstly, in favour of S1P rather than sphingosine being the biologically active determinant of glucose uptake is the observation that in SphK1^{-/-} RBC, there is no reduction of glucose uptake despite considerable sphingosine accumulation and in the absence of S1P generation (Fig. 2I, J, K). We have also performed experiments with FTY720 and FTY720-P (now to be found as new Fig. 4). The charm of this approach is that FTY720 is phosphorylated only by sphingosine kinase 2 that is not expressed in RBCs. This is confirmed by the complete lack of conversion of FTY720 to FTY720-P in Sphk2ko RBC. Accordingly, in Sphk2ko RBC there is no reduction of glucose uptake by FTY720 but a very well with FTY720-P.

#2.

The authors should also show the dependency of the modulation of glucose uptake on intracellular concentration of S1P in pharmacological modulations of intracellular S1P and/or erythrocytes from heterogeneous knockout of Mfsd2b.

These are also very meaningful experiments that we have gladly performed. In the figure below we show data comparing glucose uptake of RBCs from Mfsd2b^{+/+}, Mfsd2b^{+/-} and Mfsd2b^{-/-}. There is, indeed, a tendency of lower glucose uptake in RBC from heterozygous mice but the statistical effect is visible only in the complete knockout. We are also providing the S1P levels we have measured in these genotypes. As to pharmacological experiments in vivo, we would refer to the data on RBC from S1P lyase inhibited mice (Fig 2. A-C) that are confirmed by the genetic lyase knockout (Fig. 2D-F). Although science usually asks for genetic evidence after pharmacological evidence, we have done this also the other way around: we find the Sphk1^{-/-} glucose uptake data nicely corroborated with the Sphk1 inhibitor PF543 (see below).

#3. I have concerns that the extents of the modulations of intracellular S1P on those of the influences on glucose uptake are not concordant.

Fig 1A, C: As much as 15 fold increase of S1P resulted in 34 % decrease of glucose uptake.
 Fig 1B, C: Only 40% decrease of S1P restored 34 % decrease of glucose uptake.
 Fig 2D, E; Only 1.8 fold increase of S1P resulted in 26 % decrease of glucose uptake.
 Fig 2G, I: 13 fold decrease of S1P resulted in 34 % increase in glucose uptake.
 Fig 3A, C: 40-fold increase of S1P resulted in 20 % decrease in glucose uptake.

I have concerns that these data suggested that the modulations of S1P might not directly affect glucose uptake, but some indirect effects such as physical modulations of membranes by extremely modulated S1P.

We completely understand the concerns. There are two issues that should consider be considered in relation to such these calculations. We see a solid non-linear correlation between glucose uptake and S1P content in regular mouse RBC (see below) where the non-linearity (and saturability) is observed already at very moderate increases in S1P. This plot does not include the “extremely” high concentrations of *Mfsd2b*^{-/-} mice. Interestingly, however, RBC can apparently accommodate even such “extremely” (50-60-fold higher) S1P concentrations without any adverse effects on RBC number, Hb, MCH, MCHC, except a mild increase in MCV. Secondly, our FTY720 data and the *Sphk1*^{-/-} data are also devoid of extremes but clearly support physiological specificity

#4, Fig. 3D, 3F. If the mechanisms for the regulation of glucose uptake by S1P are not limited to erythrocytes, how about other organs involved in glucose metabolism, such as liver, muscle, and adipose tissues?

We don't know this but will address it in the future. Here we have limited ourselves to RBC.

#5. Fig 4E. What vehicles are used to deliver S1P or other lipids? S1P might not be resolved in solution, such as H₂O. The authors present these experiments for the evidence of S1P involvement of PP2A activity, however, these experiments are not physiological.

We have delivered S1P dissolved in respective vehicles as required by the experimental setting and always provide the vehicle control, its identity and concentration in each figure and figure legend. For sphingosine ethanol (0,1%) serves as vehicle, for S1P methanol (0,1%), EtOH (0,1%) for FTY720 and DMSO (0,1%) for FTY720-P. Neither one of these solvents had had any effect on the parameters measured. As to S1P solubility in aqueous media: it

behaves as media as a soluble amphiphile with a critical micelle concentration of 12 μ M (García-Pacios M, Collado MI, Busto JV, Sot J, Alonso A, Arrondo JL, Goñi FM. Sphingosine-1-phosphate as an amphipathic metabolite: its properties in aqueous and membrane environments. *Biophys J.* 2009 Sep 2;97(5):1398-407. doi: 10.1016/j.bpj.2009.07.001. PMID: 19720028; PMCID: PMC2749770). As we have measured increased PP2A activity in genetic models of high S1P (Mfsd2b^{-/-}) and decreased PP2A activity in genetic models of low S1P (Sphk1^{-/-}), we believe the evidence is rather solid.

#6. Please explain the meaning of Fig. 7 in the context of the manuscript in more detail. Especially the sentence of “This clearly demonstrated that high intracellular S1P protected RBC from pathological hemoglobin glycation in chronic hyperglycemia.”

We have rephrased to explain, and we have modulated particularly in respect to our new data on lipid peroxidation.

#7. I think that Fig 7A suggest that the phenomena revealed by the authors had little impact on physiological glucose metabolism.

What is the physiological significance of this finding?

We need to complement the reviewer on this question as it made us design a truly revealing experiment. The reviewer is completely correct in that there is no impact of high RBC S1P on physiological response to GTT. However, there are very relevant consequences for the extent of lipid peroxidation in RBC: As might have been expected, DIO lead to an increase in lipid peroxidation in RBC as assessed by measuring thiobarbituric acid reactive substances (TBARS). However, RBC from DIO-exposed Mfsd2b^{-/-} mice harboring high S1P levels (and low HbA1c) had considerably less lipid peroxidation (new figure 8C). This suggests that their RBC are better protected against the adverse effects of diabetes associated with lipid peroxidation.

#8. Characteristics of the subjects in Figure 6 should be provided. Why intracellular levels of S1P are modulated in the subjects of type 2 diabetes and DIO mice?

Are levels of Mfsd2B different?

Patients' characteristics provided in supplemental table 1. As to the question “why”, we believe one reason could be the higher Sphk1 activity in DIO RBC (Fig. 7F). We have also written the following: “As to the nature of mechanism behind the increased RBC S1P content in diabetes, it may well be due to the increased Sphk1 activity we have observed in diabetic RBC and perhaps to less active MFSD2B restricting efflux as the low higher intracellular pH²⁴ typical for diabetic RBC hampers its activity⁴²”.

#9. The modulations of glut1/4 mediated by intracellular S1P/PP2A pathway might influence the ability of RBC. I think this might have physiological significance, not glucose metabolism.

This is exactly the case. Thank you for enticing us to search for other physiological RBC effects hence finding the effect on lipid peroxidation.

Were the shape, erythrocyte-related laboratory parameters such as MCV, MPV, and Mdw, vulnerability of erythrocytes, and ability to carry oxygen in the erythrocytes modulated when intracellular S1P was modulated?

All the above parameters were unchanged in our pharmacological and genetic models of high or low intracellular S1P, respectively, except in *Mfsd2b*, where there is a somewhat higher MCV and discrete spherocytosis (also observed in Vu TM, Ishizu AN, Foo JC, Toh XR, Zhang F, Whee DM, Torta F, Cazenave-Gassiot A, Matsumura T, Kim S, Toh SES, Suda T, Silver DL, Wenk MR, Nguyen LN. *Mfsd2b* is essential for the sphingosine-1-phosphate export in erythrocytes and platelets. *Nature*. 2017 Oct 26;550(7677):524-528. doi: 10.1038/nature24053. Epub 2017 Oct 18. PMID: 29045386). As to p50 O2: neither we nor Vu et al. *Nature* 2017 observed any differences between wild type and *Mfsd2b*^{-/-} mice, and neither did we observe an in S1P lyse inhibited mice.

Was hemolysis observed in *Mfsd2b* KO mice reared HFD or in extremely high glucose conditions caused by the injection of streptozotocin, in which the physiological regulation of S1P-Glut axis was disturbed, as the authors described in the Discussion?

Sorry, we haven't done these experiments. Certainly, this should be followed up in the future.

Again, thank you for having us reflect and do the experiments on the physiological effects of endogenous RBC S1P increases in hyperglycemia. They have greatly increased impact and significance of the study.

REVIEWER COMMENTS

Reviewer #1 (Remarks to the Author):

The authors have attempted to address my comments by providing new data on Glut4 flow cytometry in mouse RBCs, glycolytic flux experiments, clarification of experimental details and rebuttals. Overall, the revisions do not address my main concern that the work provides interesting correlative observations without definitive novel mechanistic insights in this area which has intriguing but somewhat controversial and complex publications. Overall, I do not believe that the revisions have proven their model shown in figure 8.

Specific comments:

1. I still find the changes in glucose uptake to be very modest. The authors state statistical significance but I have concerns about whether such changes can lead to meaningful pathophysiology.
2. Lack of mechanistic information on how S1P impacts PP2A and GLUT1, 4 membrane localization. There is no data on immunofluorescence, biochemical changes, post-translational modification and how such events are related to each other. The data show correlative changes in PP2A in isolated enzyme assays and flow cytometric measurement of GLUT1 and 4 but no experiments to connect them. In this field of GLUT biology, this is rather superficial since many classic papers addressed such issues with specific antibodies, etc. Please see PMID: 18358815 and PMID: 17635959
3. Flow cytometry of GLUT1 and GLUT4 are very modest and it is hard to believe that such minor changes could cause major changes in glycation and redox alterations seen in diabetics.
4. If this mechanism is critical in diabetes, then RBC Sphk1 KO would be a better model to see if redox stress and HbA1c changes are rescued. Global Spk1 KO and Sphk2 KO should be used as controls. Mfsd2b KO may have additional complications in RBCs due to sphingolipid buildup.
5. The author's use of FTY720 and FTY720-P to address the metabolism of sphingolipid loading in RBC is not appropriate. Instead, the authors should use: D-erythro-sphingosine, D,L-threo-dihydrosphingosine, N, N-dimethylsphingosine, and L-threo-sphingosine. Please refer to classic papers in which Sphk substrate specificity was characterized (PMID: 9726979 and further papers on human and other Sphk isoforms). The biochemistry in this manuscript is weak.

6. The fact that sphingosine loading influences glucose uptake (albeit modestly) in transfected HEK293 cells in a manner altered by Mfsd2b suggests that the mechanism is not RBC specific and could be dissected / reconstituted in this cell line which is easily amenable. A concern is that HEK293 cells use insulin-stimulated glucose transport mechanism and one wonders whether this is relevant to RBCs which use insulin insensitive glucose transport.

7. Much more detailed analysis of how sphingolipids impact PP2A in RBC is needed. The okadaic acid experiments are not convincing at all. The authors explanation / rebuttal of other studies in which sphingolipids impact PP2A is also rather cursory and none of the work is cited in the paper.

Reviewer #2 (Remarks to the Author):

First of all, I would like to commend the authors for their efforts in improving their manuscript. My very 2 minor final comments are listed below:

- please, refer to the glucose tracing experiments as 1,2,3-¹³C₃-glucose. The current C¹³C₃ glucose is not the proper way to write it.

- In the interim period between the original and revised submission, an important paper came out where similar tracing experiments show an effect of exogenous and endogenous S1P on glucose uptake and metabolism - glycolysis/pentose phosphate pathway ratios - Nemkov et al. Blood Advances 2022 - <https://pubmed.ncbi.nlm.nih.gov/36469038/>

Findings in this paper could contribute to explaining the glucose uptake (and catabolism) phenotype reported in this interesting study, as well as the lipid peroxidation phenotype (as also described by Nemkov and colleagues from the Xia and D'Alessandro labs - same groups that originally reported the role of S1P in RBC metabolic reprogramming - this time within the framework of oxidant stress induced by blood storage under blood bank conditions, in normoxia and hypoxia).

Reviewer #3 (Remarks to the Author):

The authors addressed my concerns well.

I have no further comments for this paper.

I hope that they will investigate on the physiological meanings of their findings and the impacts on human diseases in the future.

Answer to REVIEWER COMMENTS

Reviewer #1 (Remarks to the Author):

The authors have attempted to address my comments by providing new data on Glut4 flow cytometry in mouse RBCs, glycolytic flux experiments, clarification of experimental details and rebuttals. Overall, the revisions do not address my main concern that the work provides interesting correlative observations without definitive novel mechanistic insights in this area which has intriguing but somewhat controversial and complex publications. Overall, I do not believe that the revisions have proven their model shown in figure 8.

Specific comments:

1. I still find the changes in glucose uptake to be very modest. The authors state statistical significance but I have concerns about whether such changes can lead to meaningful pathophysiology.

This comment is difficult to reply to. Meaningful pathophysiology is always a matter of debate and perspective. The strongest arguments we can provide in favour of meaningful pathophysiology are from our experiments in vivo and from the fact that we do not claim that the mechanism alleviates diabetes per se but rather its pathophysiological effects on RBC such as Hb glycation and lipid peroxidation:

1) High intracellular S1P in *Mfsd2b*^{-/-} RBC prevented haemoglobin glycation in a diet-induced model of hyperglycemia/insulin resistance (Fig. 8B), where HbA1c normally increases several fold (5-6 fold in the wild type). This was also the case in DOP-treated mice in the same model suggesting a phenomenon that holds true beyond genetic models.

2) Importantly, although this has no effect on hyperglycemia and glucose intolerance, it certainly decreased lipid peroxidation in RBC. In real life, it is difficult to estimate if and how much RBC lipid peroxidation contributes to diabetes but the literature concurs that it is pathophysiologically detrimental.

We have added exactly this cautionary note in the discussion on page 14. And, we have revised the model shown in figure 8.

2. Lack of mechanistic information on how S1P impacts PP2A and GLUT1, 4 membrane localization. There is no data on immunofluorescence, biochemical changes, post-translational modification and how such events are related to each other. The data show correlative changes in PP2A in isolated enzyme assays and flow cytometric measurement of GLUT1 and 4 but no experiments to connect them. In this field of GLUT biology, this is rather superficial since many classic papers addressed such issues with specific antibodies, etc. Please see PMID: 18358815 and PMID: 17635959

We have taken the reviewer's issue of post-translational modification and causal relationship of events very seriously. We have performed a series of experiments to address this, which has led to results that have much enhanced the studies mechanistic insights.

We are very familiar and admire Naomi Taylor's work (PMID: 18358815) and have used exactly the same detection tool for GLUT1 (recombinant envelope glycoprotein from human T lymphotropic virus (HTLV) fused to EGFP (H2-EGFP)^{44, 45}) to generate results. Dr. Taylor hasn't looked at anything else but GLUT1 cell surface expression by H2-EGFP. PMID: 17635959 was new to us, and, hopefully,

our newly performed Western blots on GLUT1 phosphorylation and expression will satisfy the reviewer.

Indeed, the rapid functional changes and GLUT1 localization kinetics we have observed may very well be related to rapid GLUT1 phosphorylation and dephosphorylation events that are known to affect GLUT1 cell surface translocation and glucose transport (PMID: 25982116). The most relevant phosphorylation site is Serine 226, phosphorylation of which is required for rapid increase in glucose uptake and enhanced cell surface localization of GLUT1 after stimulation with the phorbol ester 12-O-tetradecanoyl-phorbol-13-acetate (TPA) (Lee et al., Molecular Cell 2015, PMID: 25982116). However, data on physiological GLUT1 phosphorylation in normal human RBC is extremely scarce (PMID: 25982116 is the only example we were able to find), whereas a side by side analysis of GLUT1 phosphorylation and glucose uptake in RBC has never been published.

We have now done so in response to the reviewer's request. We have stimulated human RBC with TPA in the presence and absence of sphingosine and performed Western blotting with a phospho-Serine 226-specific GLUT1 antibody along with simultaneous measurements of glucose uptake rate in parallel.

We observed that TPA lead to a clear increase in GLUT1 Serine 226 phosphorylation (exactly as seen in PMID: 25982116) (Fig. 6G) that was substantially decreased by sphingosine (Fig. 6G). In addition, sphingosine alone also lead to a decrease in GLUT1 phosphorylation below baseline (Fig. 6F).

Glucose uptake behaved very similarly: TPA caused a clear increase in glucose uptake rate that was suppressed by sphingosine (Fig. 6 H). Please note that these are all paired samples allowing a paired one-way ANOVA analysis.

We have now included these data in the manuscript as GLUT1 phosphorylation state may be the important link and thank the reviewer for having us go this one step further.

PMID: 18358815 Montel-Hagen A, Kinet S, Manel N, Mongellaz C, Prohaska R, Battini JL, Delaunay J, Sitbon M, Taylor N. Erythrocyte Glut1 triggers dehydroascorbic acid uptake in mammals unable to synthesize vitamin C. Cell. 2008 Mar 21;132(6):1039-48.

PMID: 25982116 Lee EE, Ma J, Sacharidou A, Mi W, Salato VK, Nguyen N, Jiang Y, Pascual JM, North PE, Shaul PW, Mettlen M, Wang RC. A Protein Kinase C Phosphorylation Motif in GLUT1 Affects Glucose Transport and is Mutated in GLUT1 Deficiency Syndrome. Mol Cell. 2015 Jun 4;58(5):845-53.

PMID: 17635959 Blodgett DM, De Zutter JK, Levine KB, Karim P, Carruthers A. Structural basis of GLUT1 inhibition by cytoplasmic ATP. J Gen Physiol. 2007 Aug;130(2):157-68.

New Fig. 6 G, H. Western blotting for GLUT1 Serine 226 phosphorylation and 2DG uptake in human RBC treated or not with 1μM sphingosine for 20 minutes before a five-minute stimulation or not with 5nM 12-O-Tetradecanoylphorbol-13-acetate (TPA) at 37°C. Data are presented as mean±sem and tested with paired one-way ANOVA.

Western blotting Human blood was drawn from healthy volunteers and erythrocytes were isolated and prepared as described for mouse. RBC pellet (10×10^6) after the TPA experiment (see 'glucose uptake') was lysed in equal volume of NP-40 lysis buffer (1% IGEPAL, 25mM HEPES, 120mM NaCl, 1mM EDTA, pH 7.4) and 4x Laemmli sample buffer was added for ten minutes at 50°C. Samples were separated on a 4-20% Tris-gradient SDS-PAGE, transferred to a PVDF-membrane overnight and

membranes probed for phospho-Serine 226-GLUT1 (Thermo Fisher Scientific, Waltham, USA) and beta-actin (Sigma-Aldrich, St. Louis, USA).

Glucose uptake Human blood was drawn from healthy volunteers and erythrocytes were isolated and prepared as described before. Washed RBC's (100×10^6) were incubated in 1 ml zero glucose media (Tyrode, pH 7,0) and pre-incubated for 20 minutes with 1 μ M sphingosine before a five-minute stimulation with 5nM 12-O-Tetradecanoylphorbol-13-acetate (TPA) followed by 10 min incubation with 0.5 mM 2DG at 37°C. 2DG-6-phosphate (2DG6P) served as standard. Subsequently, 5 Mio RBCs were transferred per well in a white 96 well Optiplate™ (PerkinElmer, Waltham, Germany). Luminescence was measured in a CLARIOstar Plus microplate reader (BMG LABTECH GmbH, Offenburg, Germany).

3. Flow cytometry of GLUT1 and GLUT4 are very modest and it is hard to believe that such minor changes could cause major changes in glycation and redox alterations seen in diabetics.

We completely concur as neither do we believe in any major changes in overall glycation and redox state in diabetics. However, we do believe that such changes may be relevant in RBC, while irrelevant in muscle and fat as indicated by our data of unchanged hyperglycemia and GTT between Mfsd2b-/- and WT mice in the diet-induced hyperglycemia mouse model. We have now clearly stated this on page 14 to avoid any misunderstanding.

4. If this mechanism is critical in diabetes, then RBC Sphk1 KO would be a better model to see if redox stress and HbA1c changes are rescued. Global Spk1 KO and Sphk2 KO should be used as controls. Mfsd2b KO may have additional complications in RBCs due to sphingolipid buildup.

We agree and, as elaborated, are far from arguing that this mechanism is critical in diabetes in general. We merely suggest that it plays a role in RBC pathophysiology in the context of diabetes. We have now emphasized in the discussion on page 14. As for global Sphk1 and Sphk2 ko: this is an excellent suggestion, and, to the reviewer's discretion we disclose yet unpublished data that Sphk1 is more and Sphk2 less susceptible to diet-induced hyperglycemia and insulin resistance. We are currently working on this.

5. The author's use of FTY720 and FTY720-P to address the metabolism of sphingolipid loading in RBC is not appropriate. Instead, the authors should use: D-erythro-sphingosine, D,L-threo-dihydrosphingosine, N, N-dimethylsphingosine, and L-threo-sphingosine. Please refer to classic papers in which Sphk substrate specificity was characterized (PMID: 9726979 and further papers on human and other Sphk isoforms). The biochemistry in this manuscript is weak.

Yes, we agree here as well and have performed the requested experiments. We provide the data in Fig. 2L. We have observed that in contrast to D-erythro-sphingosine (that we have been throughout the study as 'sphingosine' as indicated in the methods section), neither D,L-threo-dihydrosphingosine, N, N-dimethylsphingosine or L-threo-sphingosine had any effect on glucose uptake.

Thank you for having us perform these experiments on biochemical substrate specificity. Together with the genetic data on Sphk1 and Sphk2 ko RBC and the Sgpl1vavCre as well as the pharmacological inhibition data with DOP in vivo (in Fig. 2A-K) they nicely complement the study.

New Fig. 2L. Glucose uptake rate in mouse RBC loaded with vehicle, 1 μ M D-erythro-sphingosine, 1 μ M D,L-threo-dihydrosphingosine, 1 μ M N,N-dimethylsphingosine or 1 μ M L-threo-sphingosine. Data are presented as mean \pm sem and tested with paired one-way ANOVA; $p^{**}<0.01$.

6. The fact that sphingosine loading influences glucose uptake (albeit modestly) in transfected HEK293 cells in a manner altered by Mfsd2b suggests that the mechanism is not RBC specific and could be dissected / reconstituted in this cell line which is easily amenable. A concern is that HEK293 cells use insulin-stimulated glucose transport mechanism and one wonders whether this is relevant to RBCs which use insulin insensitive glucose transport.

We completely agree, and, as detailed above, emphasize the relevance of this mechanism to RBC only. As elaborated above, the situation is probably entirely different in insulin-dependent tissues. We have abstained from any speculations.

7. Much more detailed analysis of how sphingolipids impact PP2A in RBC is needed. The okadaic acid experiments are not convincing at all. The authors explanation / rebuttal of other studies in which sphingolipids impact PP2A is also rather cursory and none of the work is cited in the paper.

We apologise. In our last response, went into great detail to read, discuss and cite all these other studies but we forgot to put our thoughts in the discussion. We have corrected this and it is, of greatest interest to put these studies in perspective. Please see page 13 first paragraph.

“...Sphingoid bases and analoga such as sphingosine, FTY720 and DMS (but not FTY720-P or S1P) have been shown to activate PP2A by binding and displacing the PP2A inhibitors SET³⁷ and ANP32A³⁸ from the holoenzyme which relieves PP2A from inhibition hence activating it. We identify another mechanism, where S1P and FTY720-P (but not sphingosine or FTY720) directly bind to and activate the catalytic subunit of PP2A in the absence of SET and ANP32A. In different cell types, these two mechanisms may be acting together or separately or even additional intermediate kinases mediators may be involved: e.g. PP2A has been shown to be activated in FTY720-treated endothelial cells³⁹, where either of these mechanisms or both may be responsible – ‘PP2A relief’ by FTY720 from SET and/or ANP32A resulting in indirect activation, and direct PP2A activation by FTY720-P after rapid phosphorylation of FTY720 by Sphk2. The only study that has described the opposite (S1P to inhibit PP2A)⁴⁰ should be interpreted with caution as the 6 μ M S1P used there on RBC clearly induced haemolysis in our hands (Supplemental Fig. 2A), and PP2A was not increased in erythrocyte-specific Sphk1 ko.”

Our detailed reflection on the topic is summarized below:

Xie T, Chen C, Peng Z, Brown BC, Reisz JA, Xu P, Zhou Z, Song A, Zhang Y, Bogdanov MV, Kellems RE, D'Alessandro A, Zhang W, Xia Y. Erythrocyte Metabolic Reprogramming by Sphingosine 1-Phosphate in Chronic Kidney Disease and Therapies. *Circ Res.* 2020 Jul 17;127(3):360-375. doi: 10.1161/CIRCRESAHA.119.316298. Here, the authors suggest that S1P inhibits PP2A

when they exogenously add 6 μ M S1P to RBC. The major problem is that 6 μ M S1P clearly induces haemolysis (see our suppl. Fig. 2A). This would, of course, reduce the amount of PP2A in the authors' subsequent antibody-based PP2A capture assay (96-well plate to which the RBC lysate is added). Hence the authors would have observed pseudo inhibition due to less PP2A being present. Also, Xie's study has performed no experiments on PP2A in any genetic or pharmacological models except an erythrocyte-specific Sphk1ko where PP2A was apparently unchanged which contradicts the inhibition.

Saddoughi SA, Gencer S, Peterson YK, Ward KE, Mukhopadhyay A, Oaks J, Bielawski J, Szulc ZM, Thomas RJ, Selvam SP, Senkal CE, Garrett-Mayer E, De Palma RM, Fedarovich D, Liu A, Habib AA, Stahelin RV, Perrotti D, Ogretmen B. Sphingosine analogue drug FTY720 targets I2PP2A/SET and mediates lung tumour suppression via activation of PP2A-RIPK1-dependent necroptosis. *EMBO Mol Med.* 2013 Jan;5(1):105-21. The authors convincingly show that FTY720 displaces/competes with SET for the PP2A holoenzyme resulting in reversal of inhibition and thus activation. Our data indicate another, different mechanism where S1P (and C6 ceramides) directly binds to and activates the PP2A catalytic subunit in the absence of SET. Habrukowich C, Han DK, Le A, Rezaul K, Pan W, Ghosh M, Li Z, Dodge-Kafka K, Jiang X, Bittman R, Hla T. Sphingosine interaction with acidic leucine-rich nuclear phosphoprotein-32A (ANP32A) regulates PP2A activity and cyclooxygenase (COX)-2 expression in human endothelial cells. *J Biol Chem.* 2010 Aug 27;285(35):26825-26831. Here, Tim Hla's group shows that another PP2A inhibitor (ANP32A) binds to biotinylated sphingosine but not to S1P. In addition, cells or pure ANP32A were treated with N,N-Dimethylsphingosine (although neither sphingosine nor S1P were tested) and PP2A activity was found to be increased. The authors conclude that PP2A is relieved from inhibition by ANP32A hence leading to its activation. This is very similar to the study on SET above. In contrast, we find S1P (but not sphingosine) to directly bind and activate PP2A in the absence of ANP32A.

Camp SM, Marciniak A, Chiang ET, Garcia AN, Bittman R, Polt R, Perez RG, Dudek SM, Garcia JGN. Sphingosine-1-phosphate receptor-independent lung endothelial cell barrier disruption induced by FTY720 regioisomers. *Pulm Circ.* 2020;10(1):10. Here, the authors show activation of PP2A by FTY720-treated endothelial cells by an unknown mechanism. This either of both mechanisms could be responsible – indirect SET- and/or ANP32A-based “PP2A relief” by FTY720 and the direct PP2A activation by FTY720-P that can be rapidly generated by HUVEC known to express Sphk1 and 2.

Reviewer #2 (Remarks to the Author):

First of all, I would like to commend the authors for their efforts in improving their manuscript.

Thank you very much. We appreciate it.

My very 2 minor final comments are listed below:

- please, refer to the glucose tracing experiments as 1,2,3-¹³C₃-glucose. The current C¹³C₃ glucose is not the proper way to write it.

We have corrected this.

- In the interim period between the original and revised submission, an important paper came out where similar tracing experiments show an effect of exogenous and endogenous S1P on glucose uptake and metabolism - glycolysis/pentose phosphate pathway ratios - Nemkov et al. *Blood Advances* 2022 - <https://pubmed.ncbi.nlm.nih.gov/36469038/>
Findings in this paper could contribute to explaining the glucose uptake (and catabolism) phenotype reported in this interesting study, as well as the lipid peroxidation phenotype (as also described by Nemkov and colleagues from the Xia and D'Alessandro labs - same groups that originally reported the role of S1P in RBC metabolic reprogramming - this time within the framework of oxidant stress induced by blood storage under blood bank conditions, in normoxia and hypoxia).

This is an excellent suggestions and we have included both the reference and discussed it pp 13-14.

Reviewer #3 (Remarks to the Author):

The authors addressed my concerns well. I have no further comments for this paper.

I hope that they will investigate on the physiological meanings of their findings and the impacts on human diseases in the future.

Thank you so much.

REVIEWER COMMENTS

Reviewer #1 (Remarks to the Author):

The authors have responded to previous comments by both provision of new data and rebuttal. Overall the manuscript is improved. The conclusions are now better supported by rigorous experimental evidence. However, I have concerns regarding the statistical significance (rather modest changes in the magnitude of changes in glucose transport). In some experiments, the number of replicates are low and the range of error is high.

Reviewer #4 (Remarks to the Author):

The manuscript by dr Thomas et al. is interesting and describes a potentially interesting phenomenon. I was tasked with a review of statistical methods and the reporting of the obtained results. Therefore, while my expertise in the biological subject matter is most likely adequate, I will refrain from evaluating the implications of the results and focus only on the data, presentation and methods used to analyse and report. On figure 1c the number of data points on the plot is 8 per group, raw data file lists 4 replicates however, which do not correspond to the values on the plots. Note that values for the Sph group are: 1023, 3369, 2397 and 2332 whereas the 8 datapoints on the plot span a range of ~1000 to 4000. Please provide raw data that would match the plot.

The differences between the groups are indeed modest and whether or not they would translate onto a biologically meaningful effect remains open to discussion.

For Panels of Figure 1 it is my understanding that the initial conditions were of vehicle or sphingosine and afterwards something (depending on experiment) was added to either group. This would mandate a paired t-test between the baseline and experiment samples but it would be improper to use a paired ANOVA for comparison of all 4 groups (as was done for figure 1e). In that case the datapoints of SPH+control IgG and SPH+Sphingomab were not paired and the comparison between those groups should not be done using a paired test. Using an unpaired test on the data provided in the source datafile would yield a p value of 0.51 in line with the Reviewer's suspicion of a modest effect. Significance achieved through the use of paired analysis may therefore be unsubstantiated.

Moreover, the paired tests are typically used on an organism or sample level experiment to mitigate interindividual variability. If the experiments depicted on figure 1 are technical replicates on a pool of RBCs and no variability other than technical would be expected, using the paired tests may be justifiable but to a more limited extent than demonstrated. It would primarily imply that the exact same cells from experiment 1 are than evaluated in all 4 experimental conditions. While this seems to be the case for the pairs vehicle vs Sph and 1%BSa vs 1%BSa vs Sph it does not hold for comparisons between all groups (Sph vs 1%BSA+Sph for instance) as demonstrated on most panels.

Furthermore, the Authors specify paired ANOVA which would be reasonable if all 4 conditions are applied in sequence on the same cell pool, but for pairwise experiments between the 4 conditions a dedicated post-hoc test (a Bonferroni-adjusted paired t-test or a more refined procedure with a step-down p correction) would be needed to ascertain between-group differences. Given the current presentation of the figures I believe that the groups 3 and 4 (Sph vs Sph+1%BSA or Sph+control IgG vs Sph+Sphingomab) should be compared with an unpaired test which will not yield a significant result.

Finally, for panels with a variable number of experiments (like panel 1e) use of a paired ANOVA was impossible as that would require datapoints for all 4 conditions whereas samples 8 and 9 have two missing values each.

This issue persists for other figures as well with some unexplained omissions of test results. Specifically, for panel 2b the difference in S1P efflux is significant in controls ($p=0.014$ paired t-test) but not marked as such. The magnitude of difference is greater in the DOP group, but the number of datapoints in the table does not match panel B. The source data file lists 9 data points, as does the figure, but the legend lists $N=5$ for both groups which is not the case. On that figure however, the use of paired tests would be justified for the $-/+$ 1%BSA conditions only. It is unclear in the figure legend whether this was the case.

On figure 3 there are similar issues – lack of significance for panel 3e between the Sph vs Sph+1%BSA which has a $p=0.013$ in a paired and 0.0032 unpaired t-test and others. Moreover, it is unclear what was the rationale behind showing only specific comparisons in this way. On panel 3f only the vehicle vs Sph is shown as significant whereas the vehicle vs Sph+1%BSA was significant as well.

On figure 4 similar issues are present with some numerical errors also identifiable. Specifically, for panel 4F the p values for vehicle vs FTY720-P is 0.0298 in a paired t-test (but is listed as <0.01), while the FTY720-P vs sam+1%BS is 0.027 and shown correctly as $p<0.05$. Comparison between vehicle and FTY720-P+1%BSA is not reported (was not significant). If ANOVA and proper post-hoc tests were used here, neither difference would retain significance. Also, I would be again reluctant to say that these conditions should be compared using a paired test.

Figures 5 and 6 properly represent the paired and unpaired analysis approached. Post-hoc tests for pairwise comparisons of groups on panels 6g and 6h are unclear. If any kind of multiple test adjustment was applied (as it typically is if ANOVA yields significance), the vehicle vs Sph on panel 6G and all comparisons on 6H would not be significant.

Figure 8 seems to report a means to adjust p values for multiple comparison testing. The legend reports “ANOVA (B-H)” - which implies Benjamini-Hochberg FDR procedure for p value adjustment in post-hoc tests, but this is not elaborated or on explained in any manner, nor is this used on other figures where such adjustments would be mandated.

In summary, the manuscript would benefit greatly from a thorough analysis of reportedly significant results, labelling them accordingly with the actual tests used to ascertain significance and reporting the results in the supplementary data in a comprehensive way. Marking of performed comparisons and/or significance on the figures should be in line with actual numerical data provided in the raw data file.

Reviewer #4 (Remarks to the Author):

The manuscript by dr Thomas et al. is interesting and describes a potentially interesting phenomenon. I was tasked with a review of statistical methods and the reporting of the obtained results. Therefore, while my expertise in the biological subject matter is most likely adequate, I will refrain from evaluating the implications of the results and focus only on the data, presentation and methods used to analyse and report.

We appreciate your careful and insightful review. I would have loved to hear your opinion on the biological subject matter as well, but alas.

First of all, we have revisited and revised the raw data file. In the past, the associate editor has asked us to submit the raw data file in at least two different versions, and now, in this third version, we did everything de novo. We have added missing values (we apologize for this), explained why values have been excluded or included for individual testing, explained the statistical tests and the reasons why we have chosen them, and, most importantly, have indicated matching samples much more clearly.

We have also added clarifications in the text, figure legends, and have now excluded 2 data points in Figure 2 (explained below).

We hope that will be deemed, hopefully, adequate by the reviewer.

Our statistician has also looked through the results as suggested by the reviewer and concurs.

On figure 1c the number of data points on the plot is 8 per group, raw data file lists 4 replicates however, which do not correspond to the values on the plots. Note that values for the Sph group are: 1023, 3369, 2397 and 2332 whereas the 8 datapoints on the plot span a range of ~1000 to 4000. Please provide raw data that would match the plot. The differences between the groups are indeed modest and whether or not they would translate onto a biologically meaningful effect remains open to discussion.

We are very sorry for this inconvenience. By mistake, not all raw files belonging to the image were shown in the last version (corrected now). There are, indeed, 8 data points per group, as correctly indicated, and the correct data are now updated in the new raw data file. Testing had been performed with the 8 data points, so nothing has changed except the raw data file.

For Panels of Figure 1 it is my understanding that the initial conditions were of vehicle or sphingosine and afterwards something (depending on experiment) was added to either group. This would mandate a paired t-test between the baseline and experiment samples but it would be improper to use a paired ANOVA for comparison of all 4 groups (as was done for figure 1e). In that case the datapoints of SPH+control IgG and SPH+Sphingomab were not paired and the comparison between those groups should not be done using a paired test. Using an unpaired test on the data provided in the source datafile would yield a p value of 0.51 in line with the Reviewer's suspicion of a modest effect. Significance achieved through the use of paired analysis may therefore be unsubstantiated. Moreover, the paired tests are typically used on an organism or sample level experiment to mitigate interindividual variability. If the experiments depicted on figure 1 are technical replicates on a pool of RBCs and no variability other than technical would be expected, using the paired tests may be justifiable but to a more limited extent than demonstrated. It would primarily imply that the exact same cells from experiment 1 are than evaluated in all 4 experimental conditions. While this seems to be the case for the pairs vehicle vs Sph and 1%BSa vs 1%BSa vs Sph it does not hold for comparisons

between all groups (Sph vs 1%BSA+Sph for instance) as demonstrated on most panels. Furthermore, the Authors specify paired ANOVA which would be reasonable if all 4 conditions are applied in sequence on the same cell pool, but for pairwise experiments between the 4 conditions a dedicated post-hoc test (a Bonferroni-adjusted paired t-test or a more refined procedure with a step-down p correction) would be needed to ascertain between-group differences. Given the current presentation of the figures I believe that the groups 3 and 4 (Sph vs Sph+1%BSA or Sph+control IgG vs Sph+Sphingomab) should be compared with an unpaired test which will not yield a significant result.

Finally, for panels with a variable number of experiments (like panel 1e) use of a paired ANOVA was impossible as that would require datapoints for all 4 conditions whereas samples 8 and 9 have two missing values each.

We realize that we should have done better in explaining how the experiments were performed and the reasons for using e.g. stack matched two-way ANOVA followed by a Tukey's multiple comparison test versus ordinary two-way ANOVA followed by Tukey's or stack matched one-way ANOVA plus Tukey's post hoc.

We apologize for this. Actually, it is important to note that every experiment was performed with an individual blood sample dedicated to this and only to this particular experiment. And, that we don't show technical replicates but biological with samples from individual people or mice. Accordingly, aliquots of the identical blood sample were distributed among four different treatments.

Also, we had decided to show the original data instead of the smoother presentation based on "calculated percent of control". This leads, of course, to larger SD and more demanding statistics. However, we believe that the inter-individual differences at baseline need to be taken into account.

And, yes, we did use a post-hoc test, we just hadn't mentioned it. We apologize for this, too. We have now specified that we have used a Tukey's multiple comparison test wherever applicable and as suggested by Graphpad Prism 9.0 in settings where all 4 conditions were compared among cells from the identical cell population. For obvious reasons we cannot do sequential treatments on the identical cell but do them on aliquots of the same cell population.

In respect to the reviewer's concerns about panels with a variable number of experiments as in panel 1e (which is actually the exception in the manuscript) we would respond in the following manner:

In panel 1e, we could not perform all experiments all conditions because of sphingomab antibody shortage. This forced us to choose what question would be most important and decided that it was to test if sphingomab (by extracting S1P from RBC) can rescue (increase) glucose uptake in Sph-treated cells. Indeed, this was the case in the first 2 experiments (where we performed the experiment with only group 3 and 4; see original data set). In the following experiments we decided to do all 4 groups and added sphingomab to Sph-untreated (control) cells as well as. Then we ran out of it.

As to the statistics: If we omitted groups 1 and 2 from all experiments (which would be correct for the question we want to answer) then a paired t-test ($n=9$) gives us a $p=0.009$. If we omitted the first 2 experiments (where only group 3 and 4 are performed) and did a regular repeated measures ANOVA over all 4 conditions, we would get a $p=0.0563$ between group 3 and 4.

Instead, we decided to use the mixed-effect model of stack matched two-way ANOVA (suggested by Graphpad Prism due to missing data points) that, we believe, is a good compromise. It reflects the statistical difference between group 3 and 4 and allows us to include groups 1 and 2.

However, we also have no objections if the reviewer was to insist on us presenting groups 3 and 4 only with the paired t-test. The scientific message would be communicated in a similarly efficient manner.

This issue persists for other figures as well with some unexplained omissions of test results. Specifically, for panel 2b the difference in S1P efflux is significant in controls ($p=0.014$ paired t-test) but not marked as such. The magnitude of difference is greater in the DOP group, but the number of data points in the table does not match panel B. The source data file lists 9 data points, as does the figure, but the legend lists $N=5$ for both groups which is not the case. On that figure however, the use of paired tests would be justified for the $-/+ 1\%$ BSA conditions only. It is unclear in the figure legend whether this was the case.

In this revised version, panels 2a and 2b have been analyzed now with stack matched two-way ANOVA (instead of the mixed-effect model of stack matched two-way ANOVA in the previous version). The “unexplained” omission (it was due to one outlier) has been dealt with in a manner that we have now excluded the whole experiment if even one treatment had led to an outlier (instead to excluding only the outlier as in the previous version). The stack matched two-way ANOVA compared each cell mean with the other cell mean in that column to test for differences caused by the treatment (sample from same organism). It also allowed to study the differences underlying the different organisms ($+/-$ DOP; different organisms).

In the revised version we did not lose any of the previously found significances. The excluded data are now marked with an * and shown in Blue in the raw data file.

Furthermore, if we followed the reviewer's suggestion and applied a paired t-test, there is, indeed, a significant difference in S1P efflux with and without BSA in the control mice as well. This may not be surprising as S1P can be extracted by BSA in a normal RBC as well. By using stack matched two-way ANOVA we wanted to emphasize the finding that more S1P can be extracted from the DOP-RBC (at the expense of not finding the difference we do in the paired t-test).

We have now added the requested comparison, and have added a remark in the figure legend on the significance in the controls if a paired t-test is used.

“The legend lists $N=5$ for both groups which is not the case.” We are very sorry for this inconvenience. Figure and source data was correct, just not the figure legend. Now the figure legend has been corrected accordingly.

On figure 3 there are similar issues – lack of significance for panel 3e between the Sph vs Sph+1%BSA which has a $p=0.013$ in a paired and 0.0032 unpaired t-test and others. Moreover, it is unclear what was the rationale behind showing only specific comparisons in this way. On panel 3f only the vehicle vs Sph is shown as significant whereas the vehicle vs Sph+1%BSA was significant as well.

The reason why the comparison between vehicle and Sph+1%BSA is not shown in any of the graphs as e.g. in Figure 3f is because we don't find it relevant – the biologically important issue is the difference between Sph and Sph+BSA in the wt (no difference) and Mfsd2b cells (major difference) as in figure 3f.

Nevertheless, the reviewer is right with their statement concerning figure 3e, and we have done as requested. Again, the issue of stack matched two-way ANOVA and paired t-test (that is also valid to bring the message over) has been indicated in the figure legend.

On figure 4 similar issues are present with some numerical errors also identifiable. Specifically, for panel 4F the p values for vehicle vs FTY720-P is 0.0298 in a paired t-test (but is listed as <0.01), while the FTY720-P vs sam+1%BS is 0.027 and shown correctly as p<0.05. Comparison between vehicle and FTY720-P+1%BSA is not reported (was not significant). If ANOVA and proper post-hoc tests were used here, neither difference would retain significance. Also, I would be again reluctant to say that these conditions should be compared using a paired test.

We have now clarified this: In panel 4F, we have performed a stack matched one-way ANOVA where each row represents matched data (individual unique blood samples) with a Tukey's multiple comparison test as post-hoc test (as recommended by Graphpad Prism). With this, the indicated significant differences are 0.0058 and 0.0478. This is based on the fact that every matched value can be traced back to an individual animal. This is now in the figure legend.

Figures 5 and 6 properly represent the paired and unpaired analysis approached. Post-hoc tests for pairwise comparisons of groups on panels 6g and 6h are unclear. If any kind of multiple test adjustment was applied (as it typically is if ANOVA yields significance), the vehicle vs Sph on panel 6G and all comparisons on 6H would not be significant.

A Tukey's multiple comparison test was performed as post-hoc test in 6g and 6h on a stack matched one-way ANOVA and the significances are shown below for the reviewer's reference (copied from from Graphpad Prism). We are happy to send the original Prism files if requested.

Figure 6g

Applied statistics: ANOVA summary								
Assume sphericity?	No							
F	21.94							
F value summary	0.006							
Unusually significant p? (< 0.001)?	Yes							
Colours: One level is red	0.9533							
H required	0.8144							
Was the matching effective?								
F	3.061							
F value summary	0.0414							
Is there significant matching p? (< 0.05)?								
Yes								
H required	0.1901							
ANOVA table								
SS	DF	MS	F (DF1, DF2)	P value				
Treatment (between columns)	74.66	3	24.89	F (3, 168) = 24.89, P = 0.0001				
Residual (within treatment)	17.47	5	3.494					
Total (overall)	17.01	15	1.134					
Total	109.2	22						
Data summary								
Number of treatments (columns)	4							
Number of subjects (rows)	6							
Number of missing values	0							
Number of families								
Number of comparisons per family	1							
Alpha	0.05							
Tukey's multiple comparison test								
Mean LPI	95.0% CI or diff.	Below threshold?	Summary	Adjusted P Value				
vehicle vs. TPA	-3.566	-6.670 to -1.057	Yes	0.0143	A-B			
vehicle vs. TPA+Sph	3.209	0.009 to 6.409	Yes	0.0237	A-C			
vehicle vs. Sph	0.743	-0.8815 to 1.467	Yes	0.0311	A-D			
TPA vs. TPA+Sph	2.213	0.8819 to 3.547	Yes	0.0062	B-C			
TPA vs. Sph	4.621	1.927 to 7.114	Yes	0.0085	B-D			
TPA+Sph vs. Sph	2.211	0.2719 to 4.249	Yes	0.0256	C-D			
Total (overall)								
Mean 1	Mean 2	Mean LPI	SE of diff.	n1	n2	t	DF	
vehicle vs. TPA	2.091	2.913	-0.845	0.2955	6	6	2.109	5
vehicle vs. TPA+Sph	2.824	3.833	-1.238	0.6571	6	6	1.937	5
vehicle vs. Sph	2.891	1.289	1.2715	0.1891	6	6	6.737	5
TPA vs. TPA+Sph	5.911	3.518	2.313	0.1977	6	6	11.701	5
TPA vs. Sph	5.919	1.289	4.621	0.2972	6	6	15.543	5
TPA+Sph vs. Sph	3.698	1.289	2.411	0.2976	6	6	8.101	5

Figure 6h

Table Analyzed	Copy of Glucose uptake rate: TPA				
Hopcroft's measure ANOVA summary	No				
Assumptions met?	Yes				
F	23.88				
P value	0.0009				
P value summary	Yes				
Statistically significant (P < 0.05)?	Yes				
Levene Greenhouse's epsilon	0.5646				
It is equal?	0.8661				
Was the multiple comparison?					
F	113.1				
P value	<0.0001				
P value summary	Yes				
Is there significant multiple (P < 0.05)?	Yes				
It is equal?	0.8346				
ANOVA table	SS	DF	MS	F (DF1, DF2)	P value
Treatment (between columns)	182047	3	60682.33	F (3, 68) = 23.88	<0.0001
Individual (between rows)	999333	4	249833.25	F (4, 12) = 113.1	<0.0001
Residual (random)	138128	12	11510.67		
Total	714908	19			
Exact summary					
Number of treatments (columns)	4				
Number of subjects (rows)	5				
Number of missing values	0				

Number of families	1							
Number of comparisons per family	6							
Alpha	0.05							
Tukey's multiple comparisons test	Mean Diff	95.00% CI of diff.	Below threshold?	Summary	Adjusted P Value			
vehicle vs. Sph	274.6	50.48 to 599.6	No	ns	A-B			
vehicle vs. Sph + TPA	29.89	-378.4 to 426.2	No	ns	A-C			
vehicle vs. TPA	-355.4	-633.4 to -67.4	Yes	**	A-D			
Sph vs. Sph + TPA	245.7	37.4 to 454.0	Yes	**	B-C			
Sph vs. TPA	-634.0	-868.3 to -399.6	Yes	**	B-D			
Sph + TPA vs. TPA	-388.3	-713.7 to -62.87	Yes	**	C-D			
Total details	Mean 1	Mean 2	Mean Diff.	SE of diff.	n1	n2	q	DF
vehicle vs. Sph	3006	2732	274.6	78.85	5	5	4.863	4
vehicle vs. Sph + TPA	3006	2978	29.89	160.1	5	5	0.4694	4
vehicle vs. TPA	3006	3366	-359.4	67.30	5	5	7.552	4
Sph vs. Sph + TPA	2732	2978	245.7	31.69	5	5	10.86	4
Sph vs. TPA	2732	3366	-634.0	57.26	5	5	15.58	4
Sph + TPA vs. TPA	2978	3366	-388.3	78.84	5	5	6.869	4

Figure 8 seems to report a means to adjust p values for multiple comparison testing. The legend reports “ANOVA (B-H)” - which implies Benjamini-Hochberg FDR procedure for p value adjustment in post-hoc tests, but this is not elaborated or on explained in any manner, nor is this used on other figures where such adjustments would be mandated.

No, no, these are ordinary two-way ANOVA tests followed by Tukey's multiple comparison test. Here are the statistics from Graphpad Prism e.g. for panel 8c.

Two-way ANOVA		Ordinary		Alpha		
		0,05				
Source of Variation	% of total variation	P value	P value summary	Significant?		
Interaction	3,742	0,0169	*	Yes		
Row Factor	5,620	0,0057	**	Yes		
Column Factor	87,21	<0,0001	****	Yes		
ANOVA table		SS (Type III)	DF	MS	F (DFn, DFd)	P value
Interaction	0,01722	1	0,01722	F (1, 10) = 8,186	P=0,0169	
Row Factor	0,02587	1	0,02587	F (1, 10) = 12,30	P=0,0057	
Column Factor	0,4013	1	0,4013	F (1, 10) = 190,8	P<0,0001	
Residual	0,02104	10	0,002104			
Difference between column means						
Predicted (LS) mean of normal chow	0,04854					
Predicted (LS) mean of DIO	0,3907					
Difference between predicted means	-0,3421					
SE of difference	0,02477					
95% CI of difference	-0,3973 to -0,2869					
Difference between row means						
Predicted (LS) mean of Mfsd2b ^{+/+}	0,2630					
Predicted (LS) mean of Mfsd2b ^{-/-}	0,1762					
Difference between predicted means	0,08686					
SE of difference	0,02477					
95% CI of difference	0,03166 to 0,1420					
Interaction CI						
Mean diff, A1 - B1	-0,4130					
Mean diff, A2 - B2	-0,2713					
(A1 - B1) - (A2 - B2)	-0,1417					
95% CI of difference	-0,2521 to -0,03135					
(B1 - A1) - (B2 - A2)	0,1417					
95% CI of difference	0,03135 to 0,2521					
Data summary						
Number of columns (Column Factor)	2					
Number of rows (Row Factor)	2					
Number of values	14					

Compare cell means regardless of rows and columns

Number of families		1			
Number of comparisons per family		6			
Alpha		0,05			
Tukey's multiple comparisons test	Predicted (LS) mean diff,	95,00% CI of diff,	Below threshold?	Summary	Adjusted P Value
Mfsd2b ^{+/+} :normal chow vs. Mfsd2b ^{+/+} :DIO	-0,4130	-0,5202 to -0,3058	Yes	****	<0,0001
Mfsd2b ^{+/+} :normal chow vs. Mfsd2b ^{-/-} :normal chow	0,01599	-0,08324 to 0,1152	No	ns	0,9589
Mfsd2b ^{+/+} :normal chow vs. Mfsd2b ^{-/-} :DIO	-0,2553	-0,3625 to -0,1481	Yes	***	0,0001
Mfsd2b ^{+/+} :DIO vs. Mfsd2b ^{-/-} :normal chow	0,4290	0,3218 to 0,5362	Yes	****	<0,0001
Mfsd2b ^{+/+} :DIO vs. Mfsd2b ^{-/-} :DIO	0,1577	0,04315 to 0,2723	Yes	**	0,0082
Mfsd2b ^{-/-} :normal chow vs. Mfsd2b ^{-/-} :DIO	-0,2713	-0,3784 to -0,1641	Yes	****	<0,0001

Test details	Predicted (LS) mean 1	Predicted (LS) mean 2	Predicted (LS) mean diff,	SE of diff,	N1	N2	q	DF
Mfsd2b ^{+/+} :normal chow vs. Mfsd2b ^{+/+} :DIO	0,05654	0,4695	-0,4130	0,03503	4	3	16,67	10,00
Mfsd2b ^{+/+} :normal chow vs. Mfsd2b ^{-/-} :normal chow	0,05654	0,04055	0,01599	0,03243	4	4	0,6971	10,00
Mfsd2b ^{+/+} :normal chow vs. Mfsd2b ^{-/-} :DIO	0,05654	0,3118	-0,2553	0,03503	4	3	10,31	10,00
Mfsd2b ^{+/+} :DIO vs. Mfsd2b ^{-/-} :normal chow	0,4695	0,04055	0,4290	0,03503	3	4	17,32	10,00
Mfsd2b ^{+/+} :DIO vs. Mfsd2b ^{-/-} :DIO	0,4695	0,3118	0,1577	0,03745	3	3	5,956	10,00
Mfsd2b ^{-/-} :normal chow vs. Mfsd2b ^{-/-} :DIO	0,04055	0,3118	-0,2713	0,03503	4	3	10,95	10,00

In summary, the manuscript would benefit greatly from a thorough analysis of reportedly significant results, labelling them accordingly with the actual tests used to ascertain significance and reporting the results in the supplementary data in a comprehensive way. Marking of performed comparisons and/or significance on the figures should be in line with actual numerical data provided in the raw data file.

Thank you again for the insightful and detailed review. We hope to have resolved the issues in a satisfactory manner. The revised raw data file, manuscript and figures are included.

Reviewer #4 (Remarks to the Author):

The Authors respond to my review meticulously and in full. I greatly appreciate the detailed explanations and answers which show diligence and awareness of the encountered statistical issues and concept of solving them in a statistically adequate way.

Discrepancies between raw data and figures have been addressed throughout. I agree with the Authors' explanations and justification of using certain tests (eg. Stack matched two-way ANOVA).

The expanded information on which datapoints were biological or technical replicates and the added descriptions in figure legends clarify the rationale behind the way the results were presented. The raw data provided with the manuscript matches with the figures and, together with the expanded clarifications on the methods used in the analysis holds true and allows for replication of the experiments by other researchers.

Reply to the reviewer

Thank you very much.